# Genotoxic stress and viral infection induce transient expression of APOBEC3A and pro-inflammatory genes through two distinct pathways

Sunwoo Oh [1,2,6], Elodie Bournique [1,2,6], Danae Bowen[1,2], Pégah Jalili [1,2], Ambrocio Sanchez [1,2], Ian Ward[1,2], Alexandra Dananberg[3], Lavanya Manjunath [1,2], Genevieve P. Tran[4], Bert L. Semler [4], John Maciejowski [3], Marcus Seldin[1,2] & Rémi Buisson [1,2,5✉]

APOBEC3A is a cytidine deaminase driving mutagenesis in tumors. While APOBEC3A-induced mutations are common, APOBEC3A expression is rarely detected in cancer cells. This discrepancy suggests a tightly controlled process to regulate episodic APOBEC3A expression in tumors. In this study, we find that both viral infection and genotoxic stress transiently up-regulate APOBEC3A and pro-inflammatory genes using two distinct mechanisms. First, we demonstrate that STAT2 promotes APOBEC3A expression in response to foreign nucleic acid via a RIG-I, MAVS, IRF3, and IFN-mediated signaling pathway. Second, we show that DNA damage and DNA replication stress trigger a NF-κB (p65/IkBα)-dependent response to induce expression of APOBEC3A and other innate immune genes, independently of DNA or RNA sensing pattern recognition receptors and the IFN-signaling response. These results not only reveal the mechanisms by which tumors could episodically up-regulate APOBEC3A but also highlight an alternative route to stimulate the immune response after DNA damage independently of cGAS/STING or RIG-I/MAVS.

[1] Department of Biological Chemistry, School of Medicine, University of California Irvine, Irvine, California, USA. [2] Center for Epigenetics and Metabolism, Chao Family Comprehensive Cancer Center, University of California Irvine, Irvine, California, USA. [3] Molecular Biology Program, Sloan Kettering Institute, Memorial Sloan Kettering Cancer Center, New York, New York, USA. [4] Department of Microbiology and Molecular Genetics, UCI Center for Virus Research, School of Medicine, University of California Irvine, Irvine, California, USA. [5] Department of Pharmaceutical Sciences, School of Pharmacy and Pharmaceutical Sciences, University of California Irvine, Irvine, California, USA. [6] These authors contributed equally: Sunwoo Oh, Elodie Bournique. ✉email: rbuisson@uci.edu

The Apolipoprotein B mRNA-editing enzyme catalytic polypeptide-like (APOBEC) cytidine deaminases are interferon (IFN)-inducible antiviral factors, which are part of our innate immune system that counteract various DNA or RNA viruses, retroviruses, and retrotransposons, to protect the integrity of cells[1–3]. APOBEC enzymes promote the deamination of cytosine to uracil on single-stranded DNA or RNA, to generate mutations in virus genomes and inhibit their replication. Therefore, many types of viruses accumulate APOBEC-driven hypermutations in their genomes such as human immunodeficiency virus type 1[4,5], hepatitis B virus (HBV)[6], or Epstein–Barr Herpes virus (EBV)[7]. More recently, these hypermutations have been detected in Rubella virus and the SARS-CoV-2 (severe acute respiratory syndrome coronavirus 2) virus genome[8,9]. Despite the clear impact of APOBECs in viral response, mechanisms by which viral infection triggers their expression are still poorly understood.

In addition to their function in virus clearance, APOBEC proteins are one of the most predominant causes of genomic mutations detected in patients' tumors[10]. Recent cancer-focused genomic studies identified that ~15% of overall sequenced tumors contain APOBEC-induced mutations and 50% or more in certain types of cancer such as breast, lung, cervical, and head and neck cancer[10–13]. APOBEC3A (A3A) and APOBEC3B (A3B), 2 of 11 APOBEC family members, are the main source of APOBEC mutational signatures[13–18]. Both A3A and A3B localize in the nucleus and have direct access to genomic DNA, thus promoting mutations, kataegis, and DNA damage[13–28]. Given their ability to rewrite genomic information, A3A and A3B are major drivers of mutations that in-turn, increase diversity in tumors and thereby promote disease progression and resistance to therapies[29–31].

Despite A3A and A3B both targeting the TpC motifs on single-stranded DNA to promote the deamination of cytosine to uracil, A3A and A3B show strong preference for different types of substrates within cells. Using a yeast model, Gordenin and colleagues[32] have demonstrated that A3A favors cytidine deamination on the YTCA sequence and A3B prefers the RTCA motif (Y is a pyrimidine and R is a purine). In addition, our laboratory analyzed recurrent APOBEC mutations in thousands of cancer patients and identified that A3A but not A3B preferentially targets specific DNA stem-loop structures in genomes of tumor cells[13] and this structure preference can override A3A predilection for TpC sequence[33]. Remarkably, A3A also displays RNA-editing activity and akin to its preference for DNA stem loop, A3A also recognizes RNA stem-loop structures in a sequence-specific manner[11,34,35]. As a result of these differential preferences and activities, tumors dominated by A3A or A3B mutations can be identified based on a pattern of distinct mutational signatures. The A3A signature is predominant in cervical, head and neck, bladder, thyroid, breast, endometrial, and lung cancers, whereas A3B mutations are mostly present in breast and kidney cancers and sarcoma[11,20,32].

In contrast to mutations in DNA, RNA mutations are transient and cannot be inherited through genome duplication during cell divisions and, as a result, will disappear quickly from the cells. Therefore, monitoring A3A RNA-editing activity presents a unique opportunity to accurately measure the acute ongoing activity of A3A. Consistently, A3A mRNA levels in patient tumor samples strongly correlate with A3A RNA mutations but not with A3A-induced DNA mutations[11]. This discrepancy between A3A expression and mutational signatures in tumors may be caused by episodic APOBEC mutagenesis[12,36]. These recent studies suggest that A3A is transiently expressed in cancer cells to generate DNA mutations but not necessarily when the tumors are collected and sequenced, thus explaining the poor correlation between A3A expression and its mutational signature[11,12]. However, the specific stressors responsible for the temporary surge of A3A expression in cancer cells and mechanisms regulating A3A expression in tumors are still not known.

In this study, we identify two independent mechanisms regulating A3A expression in cells. We find that both the activation of RNA-associated molecular pattern recognition and genotoxic stress cause a strong and transient upregulation of A3A and other pro-inflammatory genes. We demonstrate that the stimulation of the transcription factor STAT2 by a retinoic acid-inducible gene I (RIG-I), mitochondrial antiviral-signaling protein (MAVS), interferon regulatory factor 3 (IRF3), and IFN-dependent immune response induces A3A expression. Furthermore, we show that DNA damage and DNA replication stress also strongly promote A3A expression but, surprisingly, by an IFN-independent signaling pathway that does not require STAT2. We find that genotoxic stress regulates A3A and other pro-inflammatory genes through the activation of the p65/IkBα pathway. More importantly, A3A expression is quickly suppressed once the stress is resolved, offering a direct explanation for why A3A is rarely detected in cancer cells. We propose a dynamic means for repeating cycles of induction and suppression of A3A expression by different stressors, which drive the accumulation of clinically described A3A mutational signature over time.

## Results

**RNA pattern recognition receptors control A3A expression after viral infection.** The first line of host defense against viruses and other infectious agents consists of the activation of innate immunity through the detection of pathogen-associated molecular patterns (PAMPs) by the host pattern recognition receptors (PRRs). Key PPRs involved in PAMPs recognition are nucleic acid sensors that detect specific DNA and RNA sequences or structures present in viral genomes to trigger multiple signaling cascades, converging on the production of type I IFNs, pro-inflammatory cytokines, and chemokines. To determine which nucleic acid sensors are involved in promoting A3A expression after a viral infection, we transfected MCF10A cells (a breast immortalized cell line) and BICR6 cells (a head and neck cancer cell line) with different DNA or RNA oligonucleotides commonly used to activate the antiviral-like immune response by stimulating different PRRs (Fig. 1a and Supplementary Fig. 1A). Transfection of both MCF10A and BICR6 cells with DNA poly(deoxyadenylic-deoxythymidylic) (poly(dA:dT)), RNA polyinosinic-polycytidylic acid (poly(I:C)) (HMW: high molecular weight and LMW: low molecular weight), and 3p-hpRNA oligonucleotides induced a transient expression of A3A (Fig. 1a, b and Supplementary Fig. 1A, B). All of these oligonucleotides are known to converge in their stimulation of the cytosolic RNA sensor RIG-I[37–40]. These observations are consistent with a previous study reporting that A3A expression correlates with the activation of both RIG-I and STING (stimulator of IFN genes) by mitochondrial cytoplasmic DNA[41]. However, causal evidence for RIG-I as essential to regulate A3A expression still remains to be demonstrated.

To determine that RIG-I is the key nucleic acid sensor responsible for promoting A3A expression under the conditions of a viral infection, we monitored the levels of A3A mRNA following transfection of the RIG-I-specific agonist 3p-hpRNA, a 5′-triphosphate hairpin RNA structure residing in the influenza A (H1N1) virus genome[37,40,42], in both MCF10A and BICR6 wild-type (WT) cells or multiple RIG-I-knockout (KO) clones. In the absence of RIG-I, 3p-hpRNA transfection failed to stimulate A3A expression (Fig. 1c and Supplementary Fig. 1C–F) and we further confirmed this result by knocking out RIG-I downstream effector MAVS (Fig. 1c and Supplementary Fig. 1G). To rule out potential

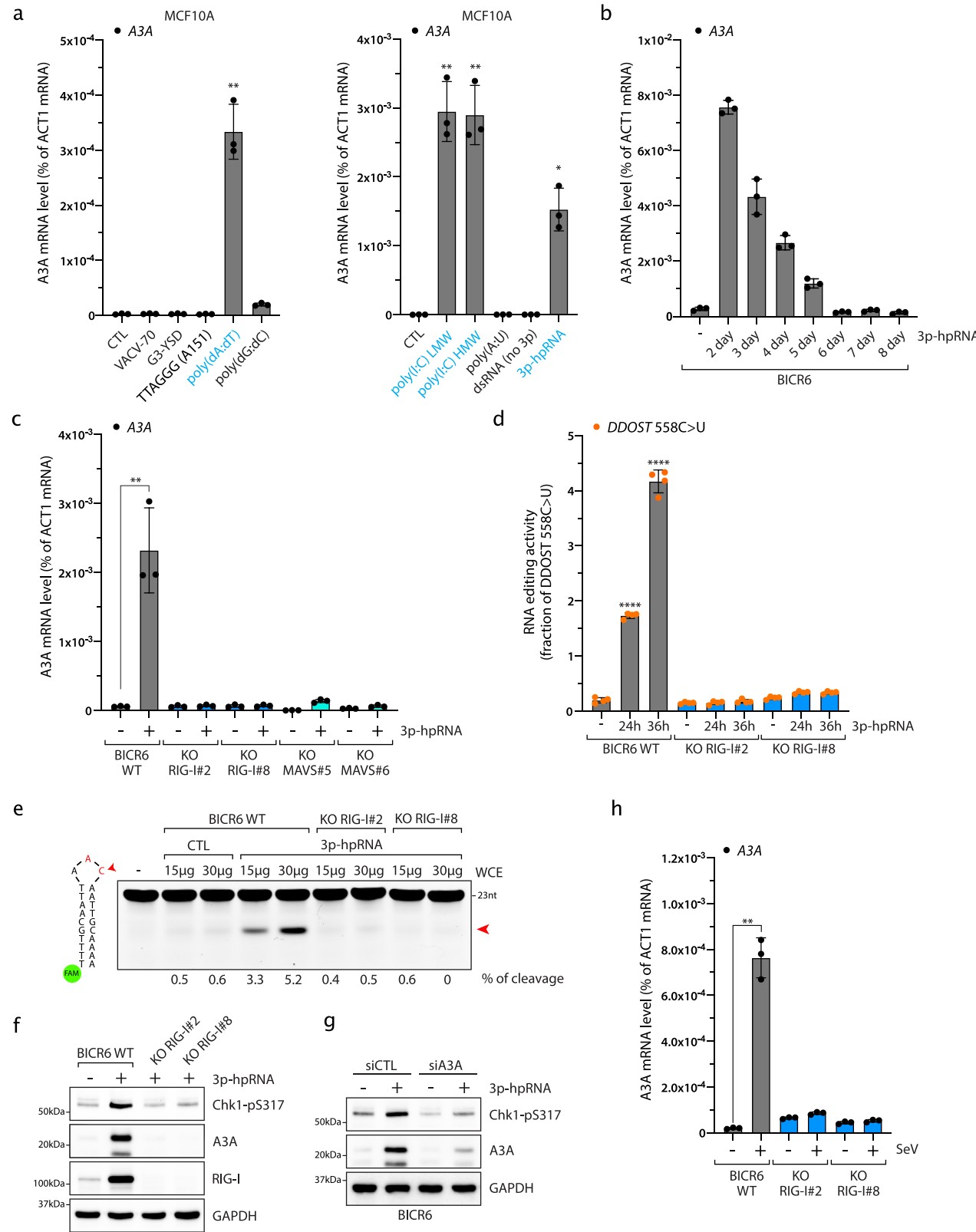

interconnection between the RIG-I/MAVS pathway and the STING pathway[41,43], we knocked out STING in MCF10A cells and repeated these experiments (Supplementary Fig. 1C). Here, the absence of STING did not suppress A3A expression after 3p-hpRNA transfection (Supplementary Fig. 1H), which is consistent with the lack of A3A expression following the transfection of VACV-70 and G3-YSD, two double-stranded DNA

oligonucleotides specifically designed to stimulate STING through the activation of IFI16 and cGAS, respectively (Fig. 1a and Supplementary Fig. 1A)[44,45]. The modest increase of A3A expression in STING KO cell lines may be the result of a hyperactivation of the RIG-I/MAVS pathway to compensate for the absence of STING in these cell lines. We recently developed a sensitive and quantitative assay to measure the ongoing activity of

**Fig. 1 Cytoplasmic double-stranded RNAs induce A3A expression by activating RIG-I and MAVS. a** Indicated DNA (200 ng/mL) or RNA (100 ng/mL) oligonucleotides were transfected in MCF10A cells. A3A mRNA level was monitored by RT-qPCR 16 h after transfection. Mean values ± SD ($n = 3$). *$P <$ 0.05, **$P < 0.01$ (two-tailed Welch $t$-test). Known RIG-I agonist oligonucleotides are labeled in blue. **b** BICR6 cells were transfected with 3p-hpRNA (40 ng/mL) and A3A mRNA level was monitored at the indicated time. Mean values ± SD ($n = 3$). **c** The A3A mRNA levels were monitored in BICR6 cells or indicated knockout cell lines 16 h after 3p-hpRNA transfection. Mean values ± SD ($n = 3$). **$P < 0.01$ (two-tailed Welch $t$-test). **d** Quantification of DDOST$^{558C>U}$ level in BICR6 WT or RIG-I KO by RNA mutation-based ddPCR assay at the indicated time following 3p-hpRNA transfection (400 ng/mL). Mean values ± SD ($n = 4$). ****$P < 0.0001$ (two-tailed Welch $t$-test). **e** A3A-deamination activity from indicated cell extracts on a DNA stem-loop substrate containing an ApC motif. **f** Indicated cell lines were transfected with 3p-hpRNA (100 ng/mL) and Chk1-pS317, A3A, and RIG-I levels were monitored by western blot 24 h after 3p-hpRNA transfection. **g** BICR6 cells were transfected with A3A siRNA for 48 h followed by 3p-hpRNA transfection (100 ng/mL). A3A and Chk1-pS317 levels were monitored in BICR6 cells 24 h after 3p-hpRNA transfection. **h** BICR6 WT and RIG-I KO cells were infected with SeV (1 MOI) for 24 h and the A3A mRNA level was quantified by RT-qPCR. Mean values ± SD ($n = 3$). **$P < 0.01$ (two-tailed Welch $t$-test). Source data are provided as a Source Data file.

A3A directly in cells by monitoring A3A RNA-editing activity at hotspots[11]. DDOST$^{558C>U}$ is the most frequent RNA hotspot mutation generated by A3A in patient tumors[11]. To test whether this acute A3A induction is strong enough to impact the cells and cause mutations, we quantified the level of DDOST$^{558C>U}$ in cells following 3p-hpRNA transfection. In WT cells, up to 4% of DDOST mRNA was edited 48 h after 3p-hpRNA transfection (Fig. 1d). This level of RNA editing is consistent with the level detected in patient tumor samples[11]. However, in the absence of RIG-I, 3p-hpRNA transfection failed to induce DDOST$^{558C>U}$ RNA mutation (Fig. 1d). We then monitored A3A deaminase activity on a DNA stem-loop substrate with an ApC motif that we previously described as a specific target of A3A[33]. Similar to A3A RNA-editing activity, cells transfected with 3p-hpRNA showed a high level of A3A enzymatic activity but not in the absence of RIG-I (Fig. 1e). Finally, A3A is known to cause replication stress[22,46,47]. To determine whether A3A expression is sufficient to induce replication stress, we monitored Chk1 phosphorylation level. Chk1 activation was detected after 3p-hpRNA transfection but absent in cells KO for RIG-I or knockdown for A3A (Fig. 1f, g), suggesting that PRR-induced A3A activity directly impacts genomic DNA and increases replication stress in the cells.

To further investigate the importance of the RIG-I/MAVs pathway in the regulation of A3A expression, we monitored A3A mRNA levels following transfection with poly(dA:dT) or Poly(I:C) oligonucleotides. Poly(dA:dT) is an A–T-rich double-stranded DNA known to activate the RIG-I/MAVs pathway through its conversion to double-stranded RNA (dsRNA) by RNA polymerase III[37,38]. Consistent with the results obtained after 3p-hpRNA transfection, the increase of A3A mRNA levels after poly(dA:dT) transfection was abrogated in RIG-I and MAVS KO cells (Supplementary Fig. 2A), and support previous observation showing that cytoplasmic mitochondrial DNA triggers A3A expression in a RNA polymerase III-dependent manner[41]. Poly(I:C) is a synthetic analog of dsRNA, structurally similar to RNA present during some viral infections and, therefore, often used to mimic the actions of foreign dsRNA. Similar to poly(dA:dT) or 3p-hpRNA, the A3A mRNA levels were strongly reduced in both RIG-I and MAVS KO cells after poly(I:C) HMW transfection (Supplementary Fig. 2B). Despite the increase in A3A mRNA level being completely abrogated in MAVS KO cells following poly(I:C) HMW transfection, A3A expression was still significantly induced in the RIG-I KO cell lines (Supplementary Fig. 2B), suggesting that another PRR was activated. In addition to RIG-I, MDA5 (Melanoma differentiation-associated protein 5) is known to detect long cytoplasmic dsRNA such as poly(I:C) HMW and to stimulate MAVS[39,48,49]. To determine whether MDA5 is involved, we knocked down MDA5 in BICR6 WT or RIG-I KO cells and monitored A3A expression. In the absence of MDA5, A3A expression induced by poly(I:C) HMW was not significantly affected in WT cells (Supplementary Fig. 2C, D),

suggesting that RIG-I activation is sufficient to stimulate A3A expression. In contrast, the incomplete suppression of A3A in the RIG-I KO cells transfected with poly(I:C) was abolished after cells were knocked down for MDA5 (Supplementary Fig. 2D). Our results demonstrate that long cytoplasmic dsRNA-induced MDA5 can act as a redundant pathway to stimulate A3A expression in the absence of RIG-I. We next asked whether direct RIG-I stimulation by a virus triggers A3A expression. Sendai virus (SeV) is a negative-sense, single-stranded RNA virus of the *Paramyxoviridae* family detected by RIG-I in infected cells[48,50]. We infected cells with SeV and monitored A3A mRNA levels over time (Supplementary Fig. 2E). Here, SeV infection strongly induced the A3A mRNA level with a maximum induction reached at 1 multiplicity of infection (MOI) (Supplementary Fig. 2F). In addition, SeV infection showed no or a modest impact on the expression of other APOBEC3 members (Supplementary Fig. 2G). Importantly, the induction of A3A expression was abrogated in BICR6 RIG-I KO cells (Fig. 1h). Taken together, these results demonstrate that the activation of the RIG-I/MDA5/MAVS pathway by foreign cytoplasmic nucleic acids promotes A3A expression.

**An IRF3-dependent IFN signaling to induce A3A expression.** Viral infection-induced RNA PRRs leads to the production of type I IFNs by triggering the nuclear localization of the transcription factor IRF3[49]. Consistently, 3p-hpRNA transfection induced the nuclear localization of IRF3 and the expression of type I IFNs (IFNα and IFNβ) in a RIG-I-dependent manner (Fig. 2a, b and Supplementary Fig. 3A–C). More importantly, A3A expression was strongly suppressed following 3p-hpRNA transfection in two independent IRF3 KO clones derived from MCF10A WT and MCF10A p53 KO, respectively (Fig. 2c and Supplementary Fig. 3D–F). To test whether A3A expression is the result of IRF3-induced IFNs production, we first transfected MCF10A WT or RIG-I KO cells with 3p-hpRNA, collected the media 24 h after transfection, and then cultured naive WT cells in the indicated conditioned media (Fig. 2d). A3A was expressed only in cells incubated with conditioned media from MCF10A WT, but not from RIG-I KO cells transfected with 3p-hpRNA (Fig. 2d), suggesting that the secretion of IFNs in the media is sufficient to stimulate A3A expression. This result is consistent with previous reports showing that purified IFNα promotes A3A expression[1,34,51–55]. To further demonstrate that IFN secretion is essential for A3A expression, we transfected MCF10A cells with 3p-hpRNA and then cultured naive IRF3 KO cells in the conditioned media. A3A expression was strongly induced by the conditioned media in both control and IRF3 KO naive cells, demonstrating that IRF3 is not directly required to drive A3A transcription but promotes A3A expression through the activation of the IFN response (Fig. 2e and Supplementary Fig. 3G).

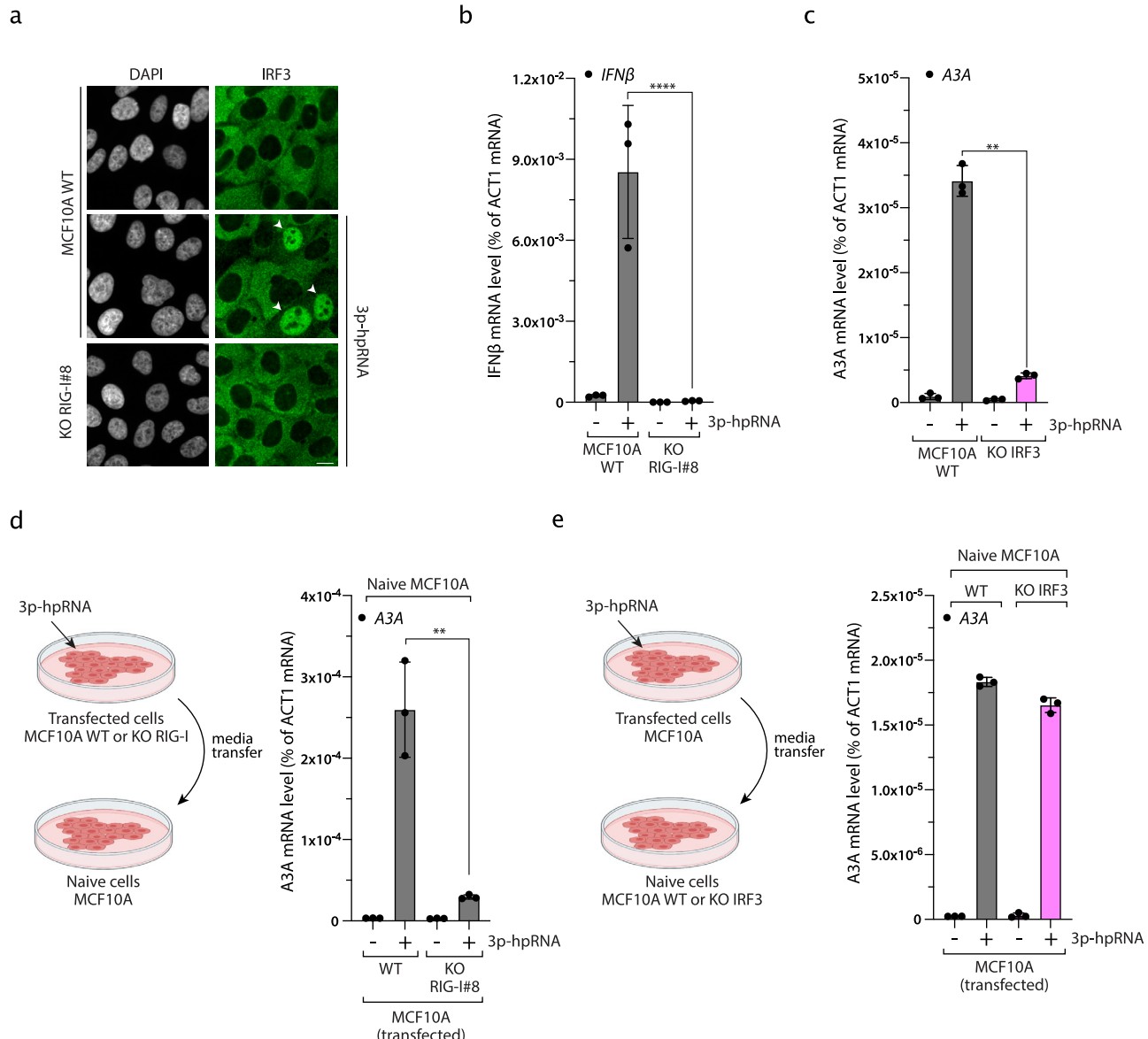

**Fig. 2 IRF3-type I interferon axis controls APOBEC3A expression. a** IRF3 nuclear localization was monitored by immunofluorescence in MCF10A WT or RIG-I KO cells 16 h after 3p-hpRNA transfection (100 ng/mL). White arrows indicate nucleus positive for IRF3. Scale bar: 10 μm. **b** Quantification of IFNβ mRNA level by RT-qPCR in MCF10A WT cells and RIG-I KO cells transfected with 100 ng/mL of 3p-hpRNA for 16 h. Mean values ± SD ($n = 3$). ****$P <$ 0.0001 (two-tailed Welch $t$-test). **c** The A3A mRNA level was monitored in the indicated cell lines transfected with 3p-hpRNA for 16 h. Mean values ± SD ($n = 3$). **$P < 0.01$ (two-tailed Welch $t$-test). **d** MCF10A WT or RIG-I KO cells were transfected with 3p-hpRNA (100 ng/mL) for 8 h followed by 16 h incubation in fresh media. Then, the conditioned media was collected, filtered, and then added to naive cells for 24 h before the quantification of A3A mRNA level by RT-qPCR. Mean values ± SD ($n = 3$). **$P < 0.01$ (two-tailed Welch $t$-test). **e** MCF10A cells were transfected with 3p-hpRNA for 8 h following by 16 h incubation in fresh media. Then, the conditioned media was collected, filtered, and then added to naive MCF10A WT or IRF3 KO cells for 24 h. The A3A mRNA levels were analyzed by RT-qPCR. Mean values ± SD ($n = 3$). Source data are provided as a Source Data file.

**Type I IFN signaling stimulates A3A expression through STAT2 after viral infection**. We then asked how the IFN response regulates A3A expression. IFN signaling activates STAT proteins through a JAKs/TYK2-mediated phosphorylation pathway. Once phosphorylated, STATs translocate to the nucleus and promote gene expression of IFN-stimulated genes (ISGs)[56]. Consistently, 3p-hpRNA transfection induced STAT1 and STAT2 phosphorylation in a RIG-I-dependent manner (Supplementary Fig. 3H). Moreover, A3A expression does not occur after 3p-hpRNA transfection in cells treated with JAK inhibitors (Fig. 3a). To determine which STATs control A3A expression, we knocked down STAT1, STAT2, or STAT3 and followed A3A mRNA levels after 3p-hpRNA transfection. Surprisingly, only in the absence of

STAT2 was the stimulation of A3A expression completely abrogated in both BICR6 and MCF10A cells (Fig. 3b and Supplementary Fig. 4A–C). This result was further confirmed in three additional model cell lines (PC-9, TPH-1, and RPE-1) (Supplementary Fig. 4D). Although STAT2 regulates many ISGs in complex with STAT1, STAT2 is also known to regulate genes by itself[57], explaining why the knockdown of STAT1 did not affect A3A levels after 3p-hpRNA transfection. However, we cannot exclude the possibility that the STAT1–STAT2 complex has a function regulating A3A levels. Regardless, in the absence of STAT1, STAT2 alone is sufficient to compensate for the loss of STAT1 and induce A3A expression. We further confirmed this result by knocking out STAT2, where both 3p-hpRNA and SeV

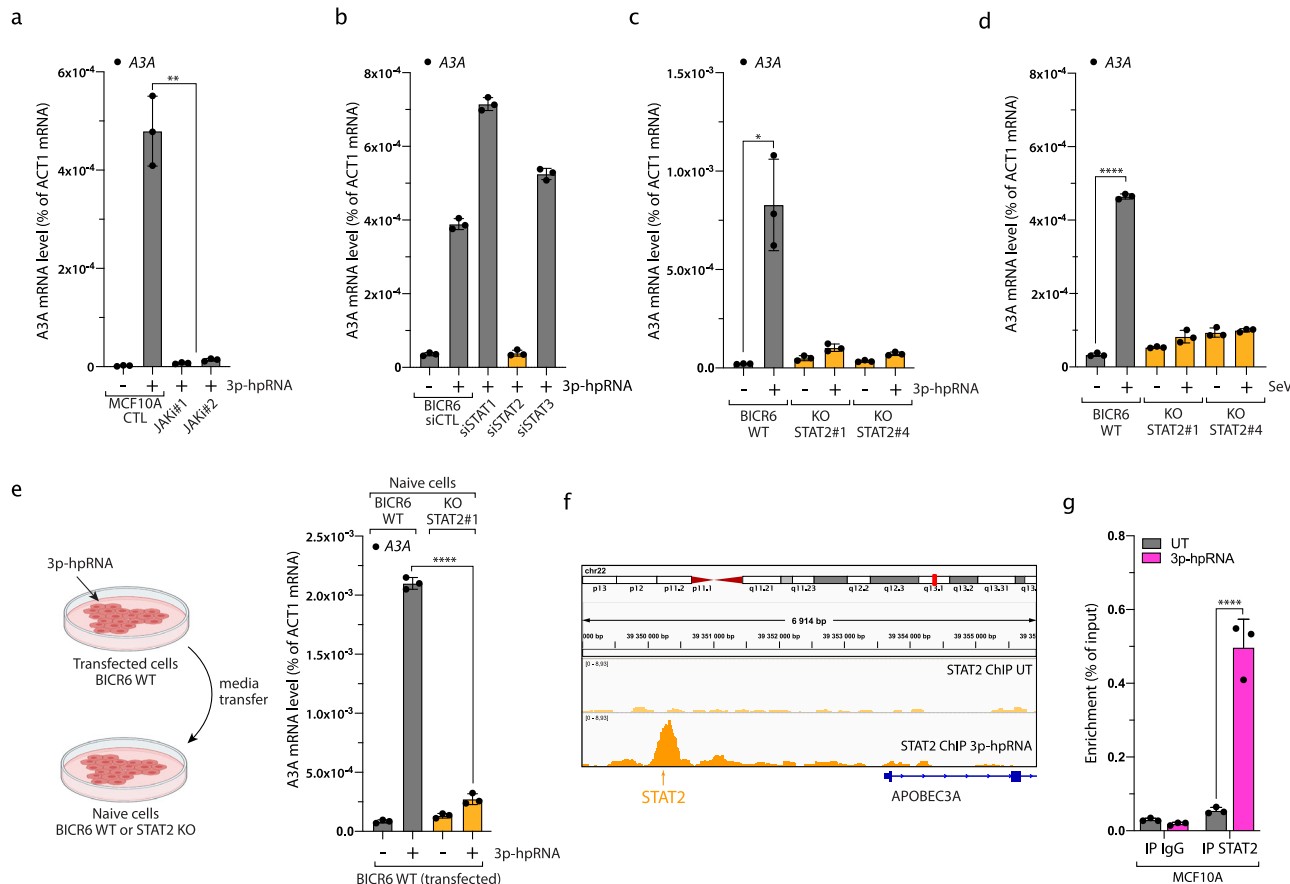

**Fig. 3 Type I IFN signaling induces A3A expression through STAT2 after viral infection. a** The A3A mRNA level was monitored in MCF10A cells 16 h after treatment with 3p-hpRNA and JAK inhibitors (JAKi #1: 2 μM Pacritinib, JAKi #2: 2 μM Ruxolitinib). Mean values ± SD ($n = 3$). $**P < 0.01$ (two-tailed Welch $t$-test). **b** BICR6 cells were transfected with indicated siRNA for 36 h followed by 3p-hpRNA transfection (100 ng/mL) for 16 h. A3A expression level was determined by RT-qPCR. Mean values ± SD ($n = 3$). **c** 3p-hpRNA was transfected in BICR6 WT or STAT2 KO cells. The mRNA level of A3A was quantified by RT-qPCR 16 h after transfection. Mean values ± SD ($n = 3$). $*P < 0.05$ (two-tailed Welch $t$-test). **d** BICR6 WT and STAT2 KO cells were infected with SeV (1 MOI) for 24 h and the A3A mRNA level was quantified by RT-qPCR. Mean values ± SD ($n = 3$). $****P < 0.0001$ (two-tailed Welch $t$-test). **e** BICR6 WT cells were transfected with 3p-hpRNA (400 ng/mL) for 8 h followed by 16 h incubation in fresh media. Then, naive BICR6 WT or STAT2 KO cells were incubated for 24 h in the conditioned media before the quantification of A3A mRNA level by RT-qPCR. Mean values ± SD ($n = 3$). $****P < 0.0001$ (two-tailed Welch $t$-test). **f** STAT2 ChIP-sequencing in MCF10A cells untreated or treated for 16 h with 3p-hpRNA. Analysis of STAT2 ChIP-sequencing data focuses on the region upstream of the A3A TSS (transcription starting site). The arrow indicates the predicted STAT2-binding site. **g** STAT2 ChIP was performed in MCF10A cells transfected with 3p-hpRNA for 16 h. STAT2 binding at the A3A promoter was determined by qPCR. The results are representative of three independent experiments and qPCR was done in triplicate. Mean values ± SD. $****P < 0.0001$ (two-tailed Welch $t$-test). Source data are provided as a Source Data file.

failed to enhance A3A expression (Fig. 3c, d and Supplementary Fig. 4E). Furthermore, we incubated BICR6 WT or STAT2 KO cells with conditioned media from BICR6 WT cells transfected with 3p-hpRNA. In the absence of STAT2, the conditioned media failed to upregulate A3A (Fig. 3e). This result suggests that STAT2-mediated A3A expression is induced by secreted IFNs. Finally, we asked whether STAT2 is directly recruited to A3A promoter. Using chromatin immunoprecipitation (ChIP) sequencing and ChIP-quantitative PCR (qPCR), we identified a STAT2-binding site upstream to the A3A transcription start site (TSS), which was only occupied when the cells are transfected with 3p-hpRNA (Fig. 3f, g). Combined, these results demonstrate that RIG-I/MAVS-induced IFN response upregulates A3A expression through STAT2 activation (Supplementary Fig. 4F).

**Genotoxic stress stimulates A3A expression**. DNA damage and DNA replication stress are additional important causes of inflammation in cancer cells[58,59]. We therefore asked whether

genotoxic stress similar to viral infection could lead to A3A upregulation in cells. We treated MCF10A and BICR6 cells with hydroxyurea (HU), which stalls and collapses replication forks, and monitored A3A mRNA levels. Both MCF10A and BICR6 cells strongly induce A3A expression after prolonged HU treatment and A3A expression occurs at a similar time as H2AX phosphorylation, suggesting that forks collapse and A3A expression are concomitant (Fig. 4a and Supplementary Fig. 5A, B). DNA replication stress activates the DNA damage checkpoint Ataxia Telangiectasia and Rad3-related (ATR) that plays a critical role in protecting cells against stalled and collapsed replication forks[60]. Inhibition of ATR alone did not affect A3A expression levels; however, ATR inhibition with two independent ATR inhibitors (ATRi) further enhanced A3A mRNA levels induced by HU treatment even at a low HU concentration, while it did not affect 3p-hpRNA-induced A3A mRNA levels (Fig. 4b and Supplementary Fig. 5C–E). Consistently, HU treatment increased A3A protein levels and A3A activity, and both are further enhanced by the inhibition of ATR (Fig. 4c–e). We then treated

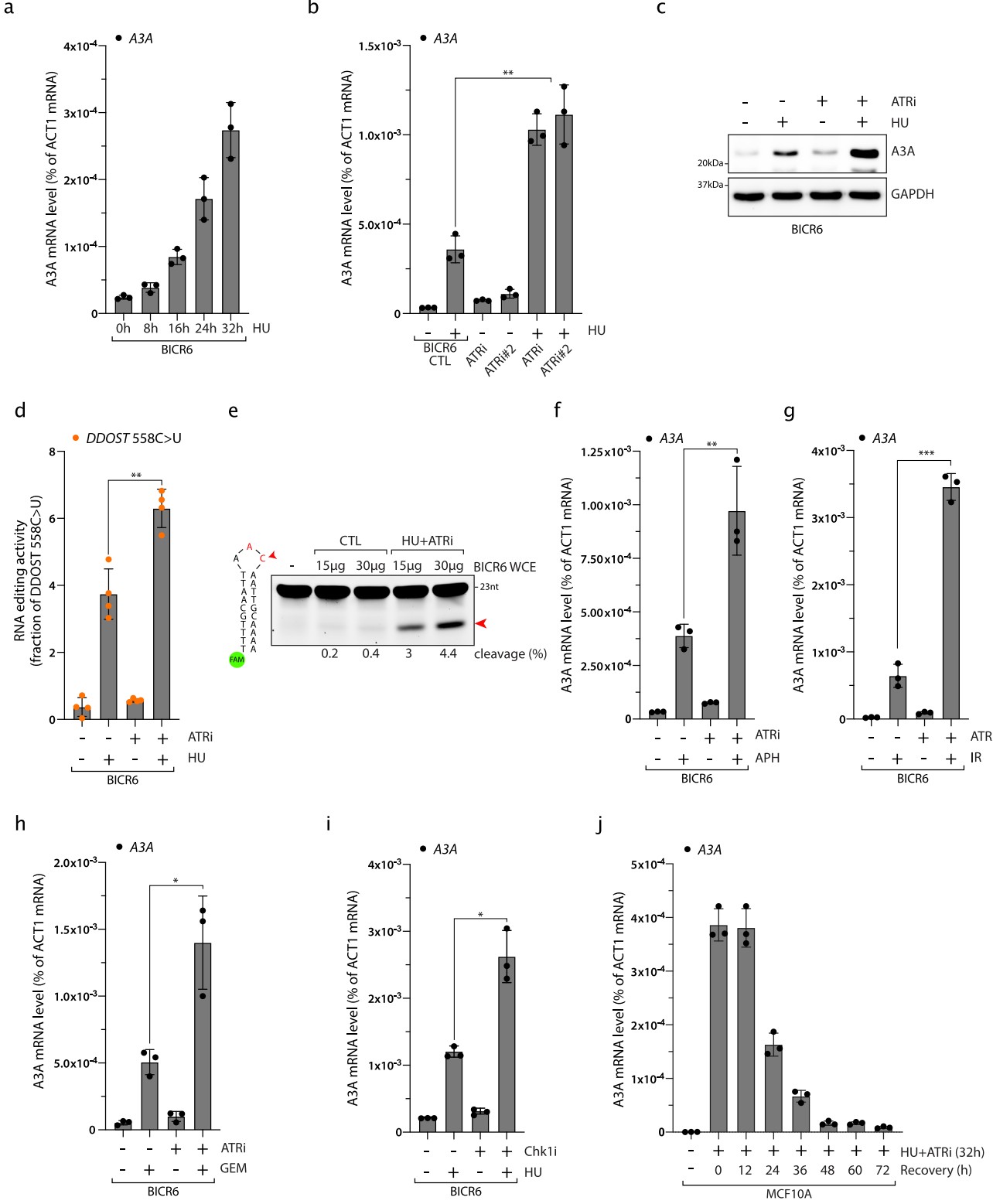

MCF10A and BICR6 cells with other types of DNA-damaging drugs in combination with ATRi. Similar to HU treatment, aphidicolin (APH), gemcitabine (GEM), and irradiation (IR) treatment also induced A3A expression that was strongly increased in the absence of ATR activity (Fig. 4f–h and Supplementary Fig. 6A–C). The inhibition of Chk1, the downstream effector of ATR in the replication stress response, also enhances A3A expression in combination with HU (Fig. 4i and

Supplementary Fig. 6D). Thus, the ATR-Chk1 signaling axis is important to prevent DNA damage-induced A3A expression in cancer cells. Finally, we observed that DNA damage-induced A3A expression is a transient process, because A3A mRNA levels declined quickly after cells are released in a drug-free medium (Fig. 4j). Together, these results imply that in addition to viral infection, DNA damage is another cause of stress leading to the upregulation of A3A expression and ATR activation is important

**Fig. 4 ATR suppresses DNA damage-induced A3A expression. a** BICR6 cells were treated with HU (2 mM) and A3A mRNA level was analyzed at the indicated times. Mean values ± SD ($n = 3$). **b** BICR6 cells were treated with HU (2 mM), ATRi (1 µM; VE-821), ATRi#2 (0.5 µM; AZD6738), or the combinations of these drugs for 32 h and the level of A3A was monitored by RT-qPCR. Mean values ± SD ($n = 3$). **$P < 0.01$ (two-tailed Welch $t$-test). **c** Level of A3A was analyzed by western blotting 48 h following the indicated treatment. **d** The level of edited $DDOST^{558C>U}$ in BICR6 cells was quantified by ddPCR assay 72 h after treatment. Mean values ± SD ($n = 4$). **$P < 0.01$ (two-tailed Welch $t$-test). **e** A3A-deamination activity from BICR6 cell extracts treated with HU + ATRi on a DNA stem-loop substrate containing an ApC motif. **f–h** BICR6 cells were treated with APH (0.5 µg/mL for 32 h), IR (10 Gy for 72 h), or GEM (0.5 µM for 32 h) in combination with ATRi (1 µM; VE-821). A3A mRNA level was monitored by RT-qPCR. Mean values ± SD ($n = 3$). *$P < 0.05$, **$P < 0.01$, ***$P < 0.001$ (two-tailed Welch $t$-test). **i** A3A expression was analyzed by RT-qPCR after 32 h of HU treatment in combination with Chk1i (2 µM). Mean values ± SD ($n = 3$). *$P < 0.05$ (two-tailed Welch $t$-test). **j** MCF10A cells were treated with HU + ATRi for 32 h and released into drug-free media for the indicated time. Mean values ± SD ($n = 3$). Source data are provided as a Source Data file.

in protecting our cells against DNA damage-induced A3A expression.

**A3A expression is induced independently of the IFN response after DNA damage.** To investigate the mechanism by which DNA damage induces A3A expression, we first asked whether the RIG-I/MAVS pathway is required to stimulate A3A expression, as found after 3p-hpRNA transfection or viral infection. We treated MCF10A and BICR6 WT or RIG-I KO cells with HU and ATRi, where, unexpectedly, the absence of RIG-I did not affect the induction of A3A mRNA level (Fig. 5a and Supplementary Fig. 7A). Similarly, we treated cells knocked out for STING and also failed to detect any suppression of A3A expression following replication stress-induced DNA damage (Fig. 5b), excluding the possibility that A3A expression was triggered by the activation of the cGAS/STING pathway through the export of small DNA fragments to the cytoplasm generated during DNA repair processes[61,62]. To understand the differential regulation of A3A expression following 3p-hpRNA transfection and HU + ATRi treatment, we directly compared both treatments side-by-side. 3p-hpRNA and HU + ATRi treatment both induce the A3A mRNA level (Fig. 5c); however, only 3p-hpRNA transfection and not HU + ATRi treatment increased IFNβ mRNA level, STAT1 and STAT2 phosphorylation, and IRF3 nuclear localization (Fig. 5c, d and Supplementary Fig. 7B, C). These results suggest that replication stress-induced DNA damage triggers A3A expression in an IFN-independent signaling manner. Consistently, whereas the A3A mRNA level was abolished in RIG-I and MAVS KO cells treated with 3p-hpRNA, A3A expression was still strongly enhanced after HU + ATRi treatment (Supplementary Fig. 7D, E). Moreover, the inhibition of JAK kinases did not affect A3A expression after HU + ATRi treatment (Supplementary Fig. 7F), further supporting that the A3A mRNA level is regulated independently of the RIG-I/MAVS pathway after DNA damage. We then asked whether cells bypass the IFN-signaling response and directly modulate STAT2 to promote A3A expression after replication stress-induced DNA damage. Similar to the absence of RIG-I, MAVS, or STING, cells knocked out for STAT2 treated with HU + ATRi did not have an impact on the A3A mRNA level compared to the WT cells, whereas STAT2 KO cells transfected with 3p-hpRNA failed to promote A3A expression (Fig. 5e and Supplementary Fig. 7G). Finally, we monitored STAT2 recruitment to A3A promoter by ChIP-qPCR. STAT2 was only present at the A3A promoter after 3p-hpRNA but not after HU + ATRi treatment (Fig. 5f), further confirming that STAT2 is not required to promote A3A expression after DNA damage. In addition, although cells treated with HU + ATRi or 3p-hpRNA induce a similar level of A3A, cells treated with both HU + ATRi and 3p-hpRNA express a higher level of A3A, suggesting that both pathways can synergistically promote A3A expression (Supplementary Fig. 7H). Together, these results demonstrated that A3A expression following replication stress-induced DNA damage is independent of the IFN response. Thus, we postulate

that cells utilize two separate mechanisms to promote A3A expression following viral infection or genotoxic stress.

Recent studies identified cGAS/STING- and RIG-I/MAVs-dependent immune responses following IR-induced genotoxic stress[62–67]. We therefore asked whether A3A expression after IR treatment was dependent on the IFN-signaling response. Comparable to HU + ATRi, IR + ATRi treatment strongly promotes A3A expression (Fig. 5g). However, only IR + ATRi but not HU + ATRi increased the IFNβ mRNA level and induced phosphorylation of STAT1 and STAT2 (Fig. 5h and Supplementary Fig. 7I). Consistent with recent reports[63,64,66,67], irradiated cells are still cycling and accumulate a high level of micronuclei caused by missegregation of chromosomes during mitosis (Supplementary Fig. 8A, B), leading to the activation of the immune response. However, after HU treatment, damaged cells are blocked in G1/S-phase and do not go through a mitosis step explaining the absence of micronuclei and IFN response after HU + ATRi treatment (Supplementary Fig. 8C, D). Surprisingly, A3A expression in RIG-I, STING, or STAT2 KO cells was not affected after IR + ATRi treatment either (Fig. 5i and Supplementary Fig. 9A, B). To explain the absence of A3A expression decrease, we then compared both 3p-hpRNA and IR + ATRi treatments. Both IFNβ expression and STAT1/2 phosphorylation after IR were modest relative to the levels induced after 3p-hpRNA transfection, whereas both treatments trigger a similar increase of A3A expression (Fig. 5j, k and Supplementary Fig. 9C), suggesting that the IFN signaling is unlikely to be responsible for A3A upregulation in irradiated cells. Consistently, STAT2 recruitment to the A3A promoter is very weak compared to recruitment in response to 3p-hpRNA transfection (Supplementary Fig. 9D). Cancer cells often deregulate cGAS-STING or other factors to counteract the detection of micronuclei and allow damaged cells to escape the immune surveillance system[67–70]. Although BICR6 cells accumulate a high level of micronuclei after IR or IR + ATRi (Supplementary Fig. 8C), they are defective for the induction of the IFN-signaling response after IR + ATRi (Supplementary Fig. 9E, F), which is most likely the result of a lack of micronuclei detection caused by the absence of STING in BICR6 cells (Supplementary Fig. 9G). Yet, BICR6 cells still strongly upregulate A3A mRNA level after IR + ATRi (Fig. 5i), which further supports that genotoxic stress induces A3A expression through an IFN-independent signaling pathway.

**Two distinct mechanisms to promote expression of pro-inflammatory genes.** To understand the differential regulation of A3A expression by the RNA-associated molecular pattern sensors and DNA damage response, we treated BICR6 cells with 3p-hpRNA or HU + ATRi and performed RNA sequencing. Transcriptomic analysis revealed the expression of a number of genes significantly increased in cells transfected with 3p-hpRNA or treated with HU + ATRi (Fig. 6a and Supplementary Data 1). As expected, all the top 3p-hpRNA-induced genes were well-known ISGs such as *MX1*, *IFIT2*, *OAS1*, *DDX60*, and *IFI44*. In

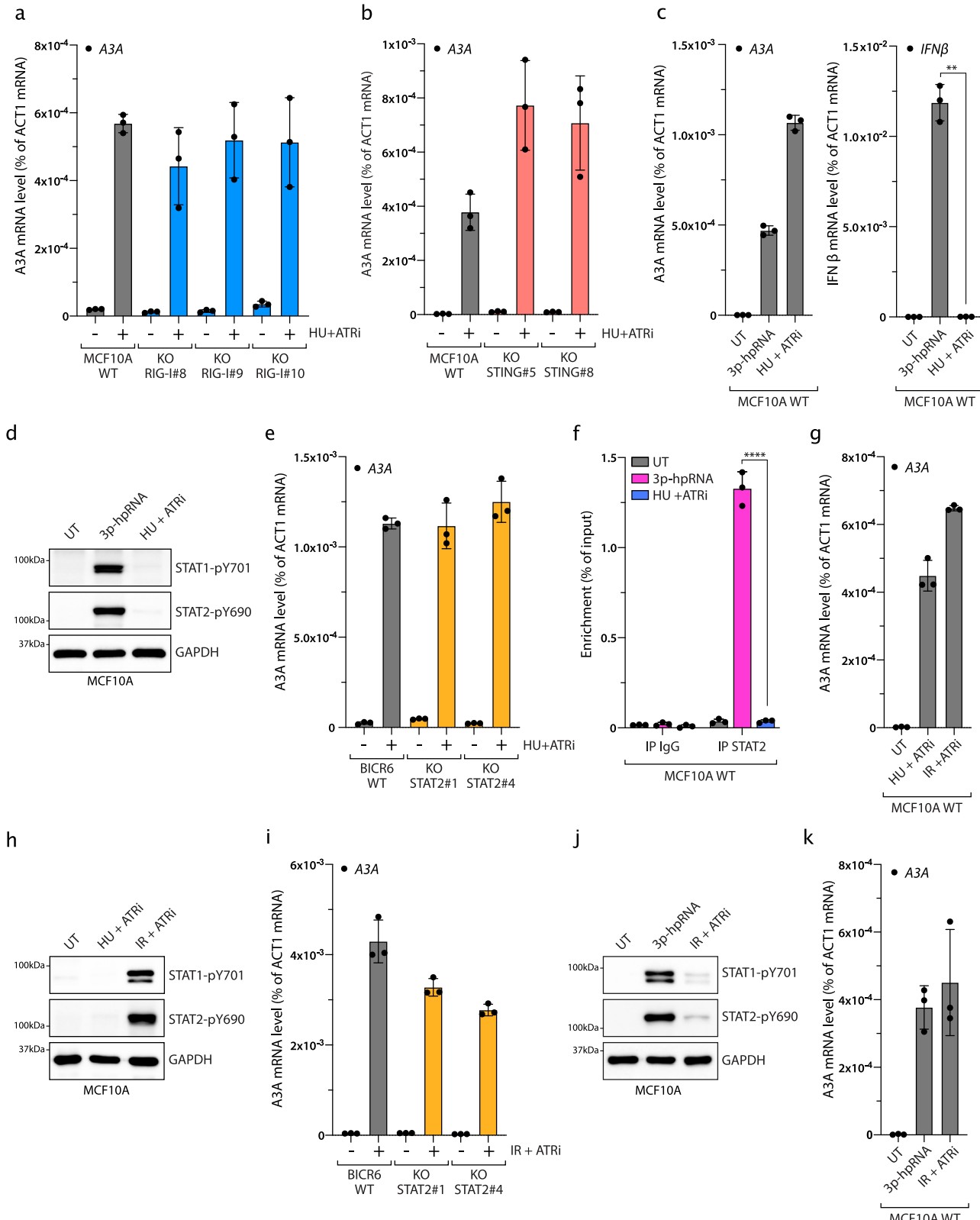

addition, we identified a sub-group of 50 genes including A3A upregulated in both 3p-hpRNA- and HU + ATRi-treated cells (Fig. 6a, b). We then asked whether these genes followed the same regulatory mechanism as A3A. We selected genes from each category (induced by 3p-hpRNA only, HU + ATRi only, or both) to serve as surrogate for pathway activation. We first confirmed the transcriptomic results by qPCR in both BICR6 and MCF10A

cells (Fig. 6c and Supplementary Fig. 10A–C), and observed that *IL-6*, *CSF2*, *CXCL2*, and *ISG20* were all upregulated following both treatments. The expression of these genes was suppressed in the absence of RIG-I or STAT2 after 3p-hpRNA transfection, whereas similar to A3A the loss of RIG-I or STAT2 did not affect their expression level after HU + ATRi treatment (Fig. 6d–f and Supplementary Fig. 10D). Together, these results indicate not

**Fig. 5 IFN-independent expression of A3A following DNA damage. a, b** A3A expression was analyzed after 32 h of HU + ATRi treatment in MCF10A WT, RIG-I KO, and STING KO cells. Mean values ± SD ($n = 3$). **c, d** MCF10A cells were treated with 3p-hpRNA or HU + ATRi for 16 and 32 h, respectively. A3A and IFNβ mRNA levels were analyzed by RT-qPCR. Mean values ± SD ($n = 3$). **$P < 0.01$ (two-tailed Welch $t$-test) (**c**), and STAT1-pY701 and STAT2-pY690 levels were monitored by western blotting (**d**). **e** BICR6 WT or STAT2 KO cells were treated with HU + ATRi for 32 h and A3A expression level was determine by RT-qPCR. Mean values ± SD ($n = 3$). **f** STAT2 ChIP was performed in MCF10A cells transfected with 3p-hpRNA for 16 h or HU + ATRi for 32 h. STAT2 binding at A3A promoter was determined by qPCR. The results are representative of three independent experiments and qPCR was done in triplicate. Mean values ± SD. ****$P < 0.0001$ (two-tailed Welch $t$-test). **g, h** MCF10A cells were treated with HU + ATRi or IR + ATRi for 32 and 72 h, respectively. The A3A mRNA level was analyzed by RT-qPCR (mean values ± SD ($n = 3$)) (**g**), and STAT1-pY701 and STAT2-pY690 levels were monitored by western blotting (**h**). **i** BICR6 WT or STAT2 KO cells were treated with IR + ATRi for 72 h and A3A expression was quantified by RT-qPCR. Mean values ± SD ($n = 3$). **j, k** MCF10A cells were treated with 3p-hpRNA or IR + ATRi for 16 and 72 h, respectively. STAT1pY701 and STAT2-pY690 levels were monitored by western blotting (**j**), and A3A levels were analyzed by RT-qPCR. Mean values ± SD ($n = 3$) (**k**). Source data are provided as a Source Data file.

only A3A but a whole subfamily of inflammatory genes are stimulated by the DNA damage response independently of the IFN-signaling response, suggesting an additional route to activate the expression of these genes.

**Expression of A3A and pro-inflammatory genes after DNA damage is dependent on p65.** Nuclear factor-κB (NF-κB) is the central transcription factor of cytokine and chemokine pro-inflammatory genes such as *IL-6*, *CCL3*, and *CXCL2*[71], all found to be upregulated after HU + ATRi treatment (Fig. 6a). Thus, we asked whether the canonical NF-κB pathway regulates A3A and other inflammatory genes after genotoxic stress. We first monitored the localization of NF-κB subunit p65 in both MCF10A and BICR6 cells, and showed a strong nuclear re-localization of p65 after HU + ATRi treatment or other types of DNA damage (Fig. 7a, b and Supplementary Fig. 11A–C). In addition, p65 nuclear localization only occurs in cells positive for γH2AX (Supplementary Fig. 11D). This result implies that the collapse of the replication forks and the formation of DNA double-strand breaks are important steps for the activation of p65 and A3A expression after replication stress. In unchallenged cells, p65 is sequestered in the cytoplasm through direct interaction with IκBα. IκBα is quickly degraded after different stresses releasing p65, which is then able to relocate into the nucleus[72]. In accordance with this model, the IκBα level decreased after HU and ATRi treatment or other types of genotoxic stress (Fig. 7c and Supplementary Fig. 11E). We then identified a canonical p65-binding sequence next to the STAT2-binding site upstream of the A3A TSS. We monitored p65 recruitment by ChIP-sequencing (ChIP-seq) and ChIP-qPCR, and observed a strong enrichment of p65 after HU + ATRi treatment at this site (Fig. 7d and Supplementary Fig. 11F). We further confirmed p65 recruitment after another type of genotoxic stress as well as by analyzing ChIP-seq data previously published for cells treated with tumor necrosis factor-α, a well-characterized activator of p65 (Supplementary Fig. 11F, G)[72]. Finally, we determined the level of DNA damage-induced gene expression in the absence of p65. Both BICR6 and MCF10A lacking p65 showed a strong defect in the stimulation of A3A expression and other pro-inflammatory genes after HU + ATRi treatment or other types of DNA damage (Fig. 7e–h and Supplementary Fig. 12A–H), and we further confirmed this result by knocking down p65 in PC-9, TPH-1, and RPE-1 cell lines (Supplementary Fig. 12I). BICR6 cell line is known to have a higher basal expression level of A3A compared to MCF10A cells[11,47] (Fig. 7i). Consistently, unchallenged BICR6 cells showed a higher intrinsic nuclear localization of p65, and knockdown or KO of p65 abrogates the basal expression of A3A (Fig. 7j, k and Supplementary Fig. 13A). On the other hand, the knockdown of IκBα enhanced the A3A level after HU + ATRi treatment (Fig. 7l and Supplementary Fig. 13B, C). Together, these results demonstrated that the canonical NF-κB pathway drives A3A expression and other pro-inflammatory genes after genotoxic stress (Fig. 8).

## Discussion
Recent cancer genomics studies identified APOBEC enzymes as one of the key drivers of mutations that increase tumor heterogeneity, metastasis, and drug resistance. A3A and A3B are mainly responsible for the APOBEC signatures detected in patient tumor samples and tumors dominated by A3A or A3B mutations can be identified[11,13,20,33]. A3A mutations are predominant in many cancer types but surprisingly A3A expression poorly correlates with A3A mutational signature, suggesting that A3A expression is transiently activated, leaving a mutational footprint even after A3A is no longer expressed[11,12]. To understand the role of A3A in tumor evolution, it is critical to characterize mechanisms controlling A3A expression level and identify initial factors triggering these mechanisms. In this study, we showed that viral infection and DNA damage are two stresses inducing an episodic surge of A3A expression using two distinct mechanisms orchestrated by the transcription factors STAT2 and p65 (Fig. 8). Remarkably, A3A expression is quickly suppressed once the stress is resolved, explaining why A3A is rarely detected in patient tumors. Repeating cycles of stimulation and suppression of A3A expression by different stressors could therefore lead to the A3A mutational signature detected in patient tumors.

In addition to A3A, A3B level is also stimulated by DNA replication stress and DNA damage[53,73,74], but our results suggest that A3A and A3B are differentially regulated. The activation of ATR prevents A3A expression, whereas ATR activity is crucial to maintain a high A3B level through an unknown mechanism[73,75]. Moreover, our study characterized the modulation of A3A expression by the canonical NF-κB transcription factor RELA (p65) after DNA damage. In contrast, A3B expression is regulated by the noncanonical NF-κB transcription factor RELB following PKCα activation[75,76]. The ability for cancer cells to increase the level of A3A through independent routes as well as the differential regulation of A3A and A3B could explain why APOBEC mutational signatures are some of the most common signatures detected in tumors with many different types of stress leading to an upregulation of A3A or A3B.

A3A is known to be induced after different types of viral infections. In this study, we explained how the RNA virus SeV triggers A3A expression by activating RIG-I. However, it is still uncertain how DNA viruses such as EBV or HBV stimulate A3A expression. Our results demonstrate that AT-rich DNA triggers A3A expression in a RIG-I-dependent manner, suggesting that a similar mechanism can be activated by DNA viruses. In accordance with this finding, EBV-positive cells induced type I IFN by a mechanism dependent on RIG-I and MAVS[37,77]. On the other hand, EBV and HBV are also known to activate the NF-κB pathway[78,79], indicating that cells may use one or both pathways to induce A3A expression after viral infection. Nevertheless, we cannot exclude that some viruses could activate an alternative pathway that still needs to be identified.

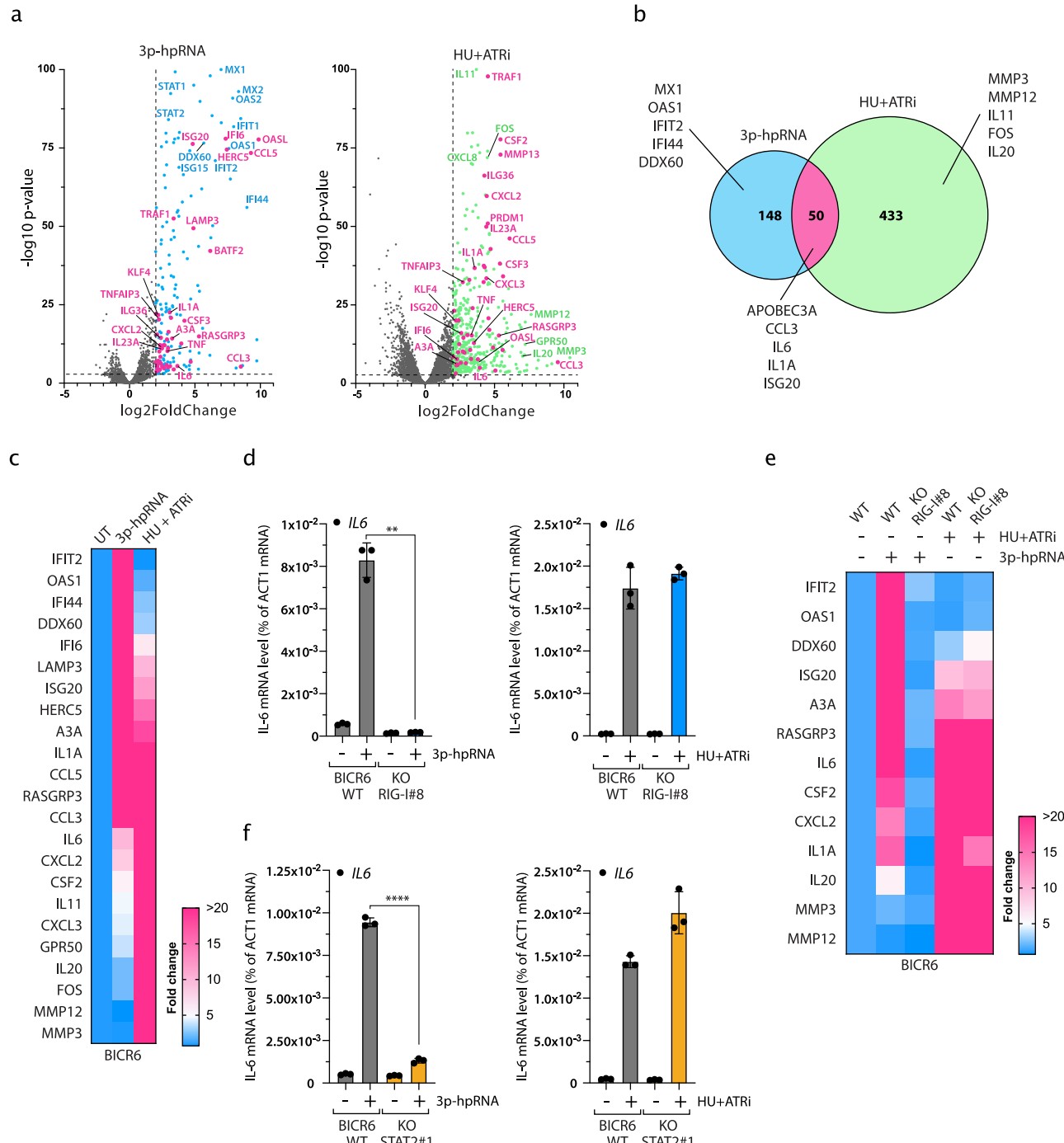

**Fig. 6 Distinct innate immune signatures mediated by IFN signaling and replication stress-induced DNA damage. a** Volcano plots of differentially expressed genes in BICR6 cells treated with 3p-hpRNA or HU + ATRi. Vertical dashed lines indicate a log2 fold-change of 2 and the horizontal dashed line indicates a *P*-value of 0.001. Data are derived from *n* = 3 biological replicates. **b** Venn diagram of significantly upregulated genes identified in **a** with a log2 (fold change) >2 and a *P*-value < 0.001. **c** Heat maps of 23 representative inflammatory genes induced after 3p-hpRNA or HU + ATRi treatment in BICR6 cells. **d** Quantification of IL-6 mRNA level by RT-qPCR in BICR6 WT and RIG-I KO cells after indicated treatment. Mean values ± SD (*n* = 3). **P < 0.01 (two-tailed Welch *t*-test). **e** Heat maps of inflammatory genes expression level in BICR6 WT or RIG-I KO treated with 3p-hpRNA or HU + ATRi. **f** IL-6 expression level was determined by RT-qPCR in BICR6 WT or STAT2 KO following indicated treatment. Mean values ± SD (*n* = 3). ****P < 0.0001 (two-tailed Welch *t*-test). Source data are provided as a Source Data file.

A3A expression is transient and quickly suppressed after the stress is resolved. Episodic expression of A3A creates a strong advantage for cancer cells compared to other mutational processes. For example, a defect of homology-directed repair or mutation in polymerase epsilon are two other common causes of mutational signature detected in tumors[10]. However, these two

mutational processes also create a vulnerability for cancer cells by increasing genomic instability, making these cells most sensitive to intrinsic stress and to chemotherapy treatment. Although A3A increases DNA replication stress and DNA double-strand breaks in cells in addition to generating mutations, episodic expression of A3A will protect cancer cells against continued A3A-induced

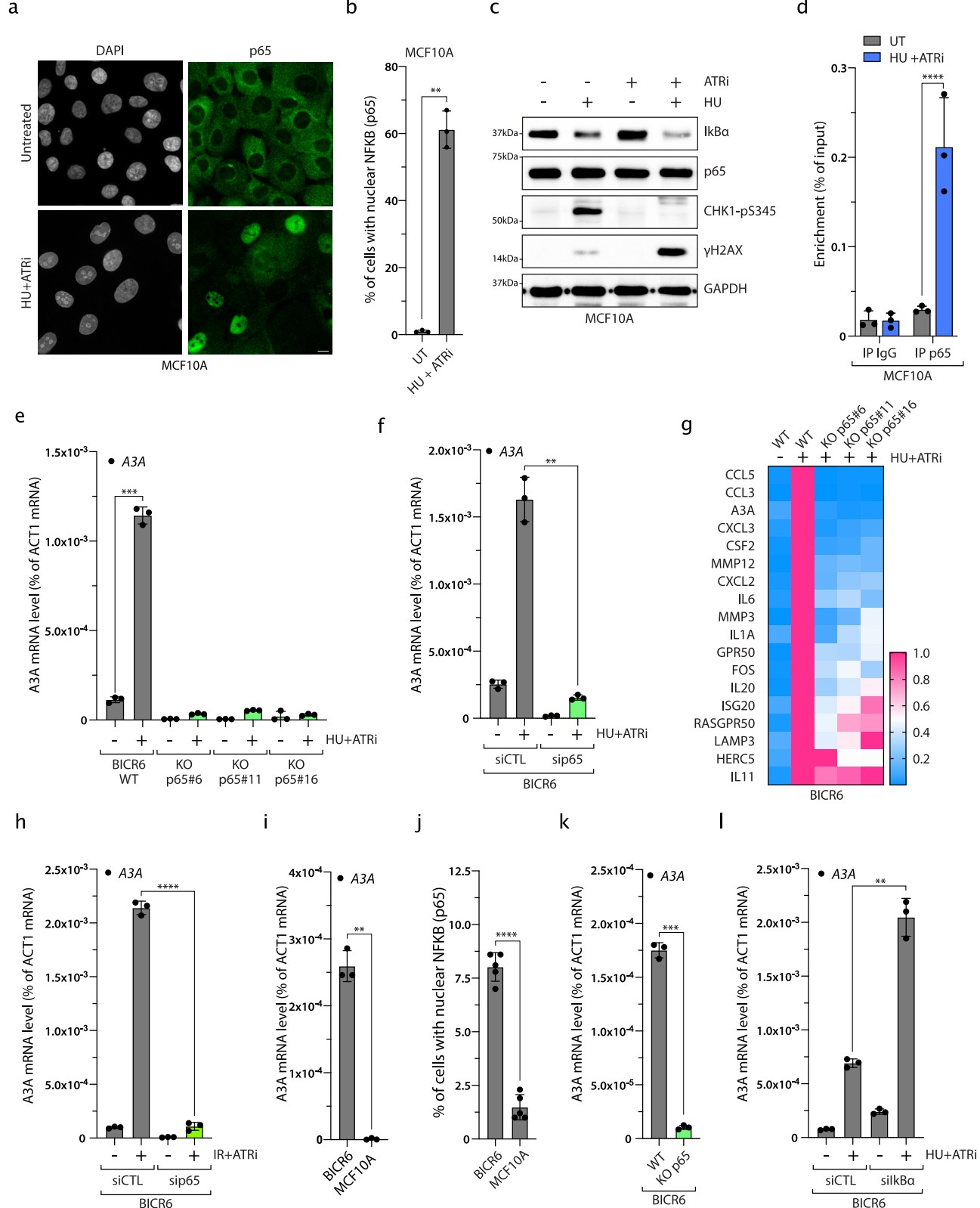

genomic instability and will render cells harder to kill using chemotherapy drugs that specifically target A3A-expressing cells. A3A-induced vulnerabilities of cancer cells can be targeted by ATRi[46,47]. Our results suggest that ATR is not only important to protect cells against A3A activity but also to prevent A3A expression. Therefore, we speculate that the use of ATRi to target cancer cells may have a dual advantage by increasing A3A levels

when used with another DNA damage chemotherapy drug and by killing A3A-expressing cells.

Understanding the mechanism of transient A3A expression in tumors is essential for the future development of therapeutic strategies to block tumor evolution and tumor heterogeneity leading to metastasis development and drug resistance. Our results suggest that chemotherapeutic drugs such as GEM or

**Fig. 7 p65 regulates A3A and pro-inflammatory gene expression after DNA damage. a** p65 localization in MCF10A cells after HU + ATRi treatment. Scale bar: 10 μm. **b** Quantification of nuclear p65 in MCF10A cells treated with HU + ATRi for 32 h. Mean values ± SD (n = 3). **P < 0.01 (two-tailed Welch t-test). **c** MCF10A cells were treated with HU and/or ATRi for 32 h and the levels of IkBα, p65, Chk1-pS345, and γH2AX were analyzed by western blotting. **d** p65 ChIP was performed in MCF10A cells treated with HU + ATRi. p65 binding at A3A promoter was determined by qPCR. The results are representative of three independent experiments and qPCR was done in triplicate. Mean values ± SD. ****P < 0.0001 (two-tailed Welch t-test). **e** A3A mRNA level quantification after HU + ATRi for 32 h in BICR6 WT or knockout for p65. Mean values ± SD (n = 3). ***P < 0.001 (two-tailed Welch t-test). **f** BICR6 cells were transfected with CTL or p65 siRNA for 40 h following by HU + ATRi treatment for 32 h. A3A expression level was determined by RT-qPCR. Mean values ± SD (n = 3). **P < 0.01 (two-tailed Welch t-test). **g** Heat maps of inflammatory genes expression level in BICR6 WT or p65 KO treated with HU + ATRi. **h** BICR6 cells were transfected with CTL or p65 siRNA for 18 h following by IR + ATRi treatment for 72 h. A3A expression level was determined by RT-qPCR. Mean values ± SD (n = 3). ****P < 0.0001 (two-tailed Welch t-test). **i** The A3A mRNA was quantified in unchallenged BICR6 and MCF10A cells. Mean values ± SD (n = 3). **P < 0.01 (two-tailed Welch t-test). **j** Quantification of nuclear p65 in the BICR6 and MCF10A cells in unchallenged conditions. Mean values ± SD (n = 3). ****P < 0.0001 (two-tailed Welch t-test). **k** The A3A mRNA was quantified in the indicated cell lines in unchallenged conditions. Mean values ± SD (n = 3). ***P < 0.001 (two-tailed Welch t-test). **i** BICR6 cells were transfected with indicated siRNA for 40 h followed by HU + ATRi treatment for 32 h. A3A expression level was determined by RT-qPCR. Mean values ± SD (n = 3). **P < 0.01 (two-tailed Welch t-test). Source data are provided as a Source Data file.

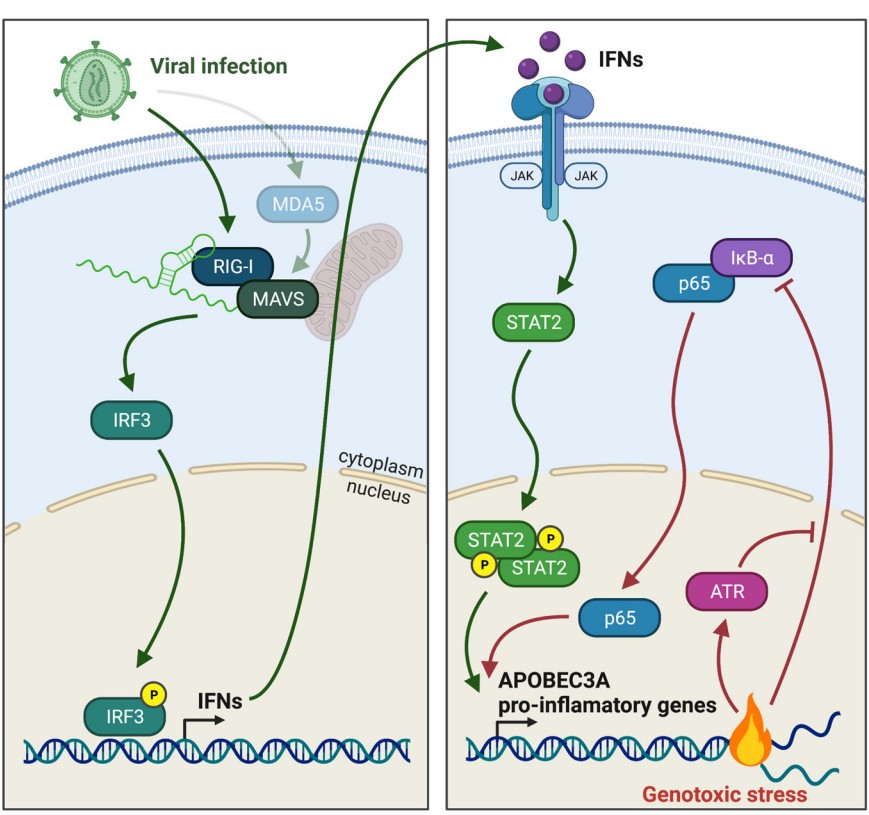

**Fig. 8 Two distinct mechanisms regulate expression of A3A and pro-inflammatory genes.** Viral infection triggers the host pattern recognition receptors RIG-I and MDA5 to promote A3A expression through the stimulation of MAVS, IRF3, and STAT2. Alternatively, genotoxic stresses lead to the activation of the canonical NF-κB pathway to transiently upregulate A3A expression.

radiation treatment may favor the emergence of resistant cancer cells by increasing A3A-mediated mutations. As an alternative to the development of inhibitors against A3A activity, suppressing A3A expression in tumors may represent a novel therapeutic strategy to prevent or delay the emergence of resistant cells. Our study suggests that inhibitors targeting the IFN response such as JAK inhibitors or the p65/IkBα pathway are potential candidates to block A3A-mediated drug resistance.

Although DNA damage-induced inflammation is well established, our study suggests that different types of genotoxic stress lead to distinct immune gene expression signatures. cGAS, RIG-I, and MDA5 detect cytosolic nucleic acids and mediate the connection between DNA damage and inflammation. Chromosome missegregation during mitosis leading to the formation of micronuclei stimulates cGAS/STING-dependent innate immune

response[63,64]. Alternatively, it has been proposed that DNA double-strand breaks that frequently occur at TA-dinucleotide repeat generate AT-rich DNA fragments[80,81]. These small pieces of DNA are then converted to dsRNA by RNA polymerase III or other mediators not yet identified to induce RIG-I/MDA5/MAVS-dependent type I IFN signaling[66,67]. Here we add a new central concept to this model where a sub-population of ISGs, including A3A, is regulated through an IFN-independent mechanism after DNA damage. This regulation requires neither cGAS/STING nor RIG-I/MAVS, but is controlled instead by the canonical NF-κB pathway. After DNA damage, nuclear localization of NF-κB directly triggers the expression of pro-inflammatory genes independently of the IFN response (Fig. 8). This activity differs from NF-κB cooperation with IRF3 to induce IFN response that has been reported after DNA double-strand

breaks[82,83], implying that NF-κB acts at several steps of the immune response after different type of DNA damage. However, the pool of expressed innate immune genes diverge regardless of whether NF-κB cooperates with IRF3 or not; thus, we propose that NF-κB coordinates different transcription response depending on its mode of activation, leading to a distinct immune gene expression signature. More importantly, our study proposes new biomarkers to monitor distinct immune gene expression signatures that may predict immunotherapy response combined with DNA damage agents.

## Methods

**Cell culture**. MCF10A cells were cultured in Dulbecco's modified Eagle medium (DMEM)/F12 supplemented with 5% horse serum, 2 ng/ml epidermal growth factor, 0.5 μg/ml hydrocortizone, 100 ng/ml cholera toxin, 10 μg/ml insulin, and 1% penicillin/streptomycin. BICR6 was maintained in DMEM/F12 GlutaMAX-I supplemented with 10% fetal bovine serum (FBS) and 1% penicillin/streptomycin. TPH-1 and PC-9 cell lines were cultured in RPMI + GlutaMAX-I supplemented with 10% FBS and 1% penicillin/streptomycin. RPE-1-hTERT (RPE-1) cell line was cultured in DMEM supplemented with 10% FBS and 1% penicillin/streptomycin. LLC-MK2 cells were cultured in Medium 199 supplemented with 1% of horse serum. Cell lines were purchased from either American Type Culture Collection (ATCC) or Sigma-Aldrich. Media from cells treated with indicated drugs was replaced with fresh medium and drugs every 24 h.

**RNA interference**. Small interfering RNA (siRNA) transfections were performed by reverse transfection with Lipofectamine RNAiMax (Thermo Fisher Scientific). siRNAs were purchased from Thermo Fisher Scientific (Silencer Select siRNA). Cells were treated with indicated drugs 32 h after siRNA transfection (4 nM). The sequences of the siRNAs used in this study are listed in Supplementary Table 1.

**CRISPR-Cas9 KO cells**. RIG-I, STING, STAT2, and p65 CRISPR-Cas9 KO cell lines were performed by transfection with Lipofectamine CRISPRMAX of True-Guide Synthetic CRISPR gRNA and TrueCut Cas9 Protein v2 according to the manufacturer's instructions (Thermo Fisher Scientific). CRISPR gene-editing efficiency was verified using GeneArt Genomic Cleavage Detection kit (A24372; Thermo Fisher Scientific). MAVS KO cell lines were performed by transfection of the pSpCas9(BB)-2A-Puro (PX459) plasmid containing MAVS gRNAs with FuGENE 6 Transfection Reagent (E2691; Promega). Sixteen hours after transfection, cells were selected with puromycin (1 μg/ml) for 2 days. IRF3 KO cell lines were performed by transfection of the pU6-(BbsI)-CBh-Cas9-T2A-mCherry plasmid containing IRF3 gRNAs with FuGENE 6 Transfection Reagent (E2691; Promega). Twenty-four hours after transfection, mCherry-expressing cells were single cell-sorted into 96-well plates. MCF10A IRF3 KO and MCF10A p53 KO cell lines was created as previously described[66,84]. For every target, three or more independent clones were generated. gRNA sequences used in this study are listed in Supplementary Table 2.

**Kinase inhibitors and chemicals**. The chemicals and concentration, if not indicated otherwise, used in this study were ATRi (1 μM VE-821, Selleckchem # S8007), ATRi #2 (0.5 μM AZD6738, Selleckchem #S7693), HU (2 mM, Sigma-Aldrich #H8627), APH (1 μg/mL Sigma-Aldrich #A0781), GEM (0.5 μM, Selleckchem #S1149), Chk1i (2 μM, CCT244747 MedChemExpress #HY-18175), JAKi #1 (2 μM Pacritinib, MedChemExpress #HY-16379), and JAKi #2 (2 μM Ruxolitinib, MedChemExpress #HY-50856). If not indicated otherwise, cells were treated with ATRi (1 μM) and HU (2 mM) for 32 h.

**Antibodies**. The antibodies used in this study are listed Supplementary Table 3.

**Oligonucleotides**. PRR ligands were purchased from InvivoGen. VACV-70 (#tlrl-vac70n), G3-YSD (#tlrl-ydna), Poly(dA:dT) (#tlrl-patn), Poly(dG:dC) (#tlrl-pgcn), ODN TTAGGG (#tlrl-ttag151), Poly(I:C)-LMW (#tlrl-piew), Poly(I:C)-HMW (#tlrl-pic), Poly(A:U) (#tlrl-pau), 5′ppp-dsRNA Control (#tlrl-3prnac), and 3p-hpRNA (#tlrl-hprna) were transfected by forward transfection with Lipofectamine 2000 (Thermo Fisher Scientific) according to the manufacturer's instructions. If not indicated otherwise, cells were treated with 3p-hpRNA (100 ng/mL) for 16 h.

**Quantitative reverse-transcription PCR**. Total RNA was extracted from cells using Quick-RNA MiniPrep Kit (Zymo Research) according to the manufacturer's instructions. Following extraction, total RNA was reverse transcribed using the High Capacity cDNA Reverse Transcription Kit (Thermo Fisher Scientific). Reverse-transcription products were analyzed by real-time qPCR using SYBR Green (PowerUp SYBR Green Master Mix, Thermo Fisher Scientific) in a QuantStudio 3 Real-Time PCR detection system (Thermo Fisher Scientific). For each sample tested, the levels of indicated mRNA were normalized to the levels of

Actin mRNA. The sequences of the PCR primers used in this study are listed in Supplementary Table 4.

**Droplet digital PCR assay**. Purified RNAs were reverse transcribed using a High Capacity cDNA Reverse Transcription Kit (Thermo Fisher Scientific). cDNA (20 ng) and indicated primers (2 μL) were added in the PCR reactions (ddPCR Supermix for Probes (No dUTP) mix from Bio-Rad) in a total of 25 μL. Then, 20 μL of the reaction mix was added to a DG8 cartridge (Bio-Rad), together with 70 μL Droplet Generation Oil for Probes (Bio-Rad), followed by the generation of droplets using a QX200 Droplet Generator (Bio-Rad). Droplets were next transferred to a 96-well plate before starting the PCR reaction in thermal cycler (C1000 Touch Thermal Cycler, Bio-Rad) under the following conditions: 5 min at 95 °C, 40 cycles of 94 °C for 30 s, 53 °C for 1 min, and then 98 °C for 10 min (ramp rate: 2 °C s⁻¹). Droplets were analyzed with the QX200 Droplet Reader (Bio-Rad) for fluorescent measurement of fluorescein amidite (FAM) and hexachloro-fluorescein (HEX) probes. Gating was performed based on positive and negative DNA oligonucleotide controls. The ddPCR data were analyzed with QuantaSoft analysis software (Bio-Rad) to obtain fractional abundances of edited RNAs. Three or more biological replicates were analyzed for each sample. DDOST primers are: DDOST Forward Sequence: ACTGAGAACCTGCTGAAG; DDOST Reverse Sequence: AAGAGGATGGGATTTAGAGA; DDOST 558C Probe Sequence: CAACCATCGTTGGGAAATC (Fluorophore: HEX), and DDOST 558T Probe Sequence: CCAACCATTGTTGGGAAATC (Fluorophore: FAM).

**Immunofluorescence**. Cells were fixed with 3% paraformaldehyde and 2% sucrose in 1× phosphate-buffered saline (PBS) for 20 min, washed twice with 1× PBS, and cells were permeabilized with a permeabilization buffer (1× PBS and 0.2% Triton X-100) for 5 min. When indicated, cells were incubated before fixation in pre-extraction buffer (10 mM PIPES pH 6.8, 100 mM NaCl, 300 mM sucrose, and 0.2% Triton X-100) for 5 min at 4 °C. Subsequently, cells were washed twice with 1× PBS and blocked in PBS-T (1× PBS and 0.05% Tween-20) containing 2% bovine serum albumin (BSA) and 10% milk for 1 h. Cells were then incubated with the primary antibody diluted in 1× PBS containing 2% BSA and 10% milk at room temperature for 2 h. Coverslips were washed three times with PBS-T before incubation (1 h) with the appropriate secondary antibodies conjugated to fluorophores (Cy3 or Alexa-488). After three washes with PBS-T, cells were stained with 4′,6-diamidino-2-phenylindole. Images were captured using a Leica DMi8 THUNDER microscope. For the quantification showed in Supplementary Fig. 11D, p65 intensity in the nucleus was subtracted from p65 cytoplasmic intensity, then was plotted against γH2AX nuclear intensity.

**Sendai virus infection**. SeV Cantell strain was purchased from ATCC (#VR-907). Viral titer was determined by plaque assay using LLC-MK2 cells. Cells were plated and infected with SeV in serum-free medium for 1 h. The LLC-MK2 cells were then overlaid and incubated with medium containing serum, 0.45% agarose, and 5 μg/ml acetylated trypsin. After 5 days, cells were fixed with trichloroacetic acid (10%) for 30 min, stained with crystal violet (0.1% crystal violet / 25% EtOH) for 5 min, and plaques counted to determine the viral titer. For virus infection, BICR6 cells were infected with SeV in serum-free medium at indicated MOI. Culture medium supplemented with serum was added 1 h post infection and infected cells were collected for quantitative reverse-transcription PCR analysis at the indicated times.

**Flow cytometry analysis**. To analyze the cell-cycle distribution, cells were pulse-labeled with 10 μM EdU for 30 min and then processed using the Click-iT EdU Alexa Fluor 647 Flow Cytometry Assay Kit according to the manufacturer's instructions (Thermo Fisher Scientific), and the DNA was stained with Propidium iodide (10 μg/mL, Sigma-Aldrich) in the presence of RNase A (50 μg/mL, Thermo Fisher Scientific). Data acquisition and analysis were performed on a NovoCyte Flow cytometer equipped with the NovoExpress software (ACEA Biosciences). Fluorescence-activated cell sorting analysis of BICR6 cells were performed with a BICR6 WT clone derived from BICR6 cell line.

**Chromatin immunoprecipitation**. ChIP experiments were performed USING the SimpleChIP Enzymatic Chromatin IP kit (Cell Signaling Technology, #9003), following the manufacturer's protocol. In brief, around 3.5 million MCF10A cells treated with indicated treatment were fixed with 1% formaldehyde for 10 min at room temperature followed by quenching with glycine. The cells were then lysed and the chromatin was fragmented by enzymatic digestion using Micrococcal Nuclease (30 min at 37 °C). IgG (2.25 μg or 3.75 μg), STAT2 (2.25 μg), or NFKB (3.75 μg) antibodies were incubated with 7.5 μg of digested and cross-linked chromatin for 16 h at 4 °C. Protein G magnetic beads were added for an additional 2 h. After immunoprecipitation, chromatin–protein complexes were eluted from protein G magnetic beads and reverse cross-linked. Eluted DNA was purified and used for qPCR and sequencing. Library generation from DNA pools and next-generation sequencing was performed on an Illumina NovaSeq platform by Novogene Corporation, Inc. For ChiP-seq analysis of STAT2 and p65, raw reads were inspected via FastQC and processed for analysis. Raw sequences were aligned to the human genome GRCh37/hg19 using STAR aligner and converted to sorted bam files via samtools (Bioproject Accession: PRJNA684601). Peak calling and

visualization were performed using HOMER tools and Integrative Genomics Viewer. The sequences of ChiP-qPCR primers used in this study are listed in Supplementary Table 5.

**RNA sequencing and data analysis**. BICR6 cells were treated with indicated treatment (three biological replicates) and total RNA was extracted using the Quick-RNA MiniPrep Kit (Zymo Cat#11–328). One microgram of total RNA for each sample was used to construct the sequencing library with the Collibri Stranded RNA Library Prep Kit Illumina$^{TM}$ with Collibri$^{TM}$ H/M/R rRNA Depletion Kit (Thermo Fisher Cat#A38110096) according to the manufacturer's instructions. Final libraries were amplified using 11 PCR cycles. Library size distributions were measured using a BioAnalyzer and quantified via qPCR. Libraries were sequenced on a HiSeq 4000 platform, set to 150 PE reads and demultiplexing (Novagene, Inc). For RNA-seq data analysis, the fastq were mapped to the human genome (GRCh38/hg38) using the STAR aligner[85], then PCR duplicates were removed using Picard tools (Bioproject Accession: PRJNA684601). The transcripts were pre-filtered such that only those with at least three samples with an FPKM (fragments per kilobase of transcript per million mapped reads) = 40 or more were included in the analysis. Then, differential expression analysis was performed using DESeq2[86], on R statistical software. To avoid $P$-values assigned as 0, based on the absolute estimation limit of DESeq2, transcripts with a DE $P$-value = 0 were assigned a new value based on a linear model estimated from distributions of correlation between $P$-value and log2FC.

**Cell extracts**. BICR6 WT or RIG-I KO cells treated with the indicated treatment were lysed in 25 mM HEPES pH 7.9, 10% glycerol, 150 mM NaCl, 0.5% Triton X-100, 1 mM EDTA, 1 mM MgCl$_2$, RNase A (0.2 μg/mL), 1 mM ZnCl$_2$, and protease inhibitors. Cell lysates were sonicated, incubated for 30 min at 4 °C, and then centrifuged 10 min at 20,000 × $g$ at 4 °C. Protein concentration of the supernatant was determined by Bradford assay (Bio-Rad).

**DNA deaminase activity assay**. Reactions (50 μL) containing 8 μL of a normalized amount of cell extracts (expressing A3A or A3B) were incubated at 37 °C during 1 h in a reaction buffer (42 μL) containing a DNA oligonucleotide (20 pmol of DNA oligonucleotide, 50 mM Tris pH 7.5, 1.5 units of uracil DNA glycosylase (NEB), RNase A (0.1 μg/mL), and 10 mM EDTA). Then, 100 nM of NaOH was added to the reaction followed by 40 min incubation at 95 °C. Finally, 50 μL of formamide was added to the reaction (50% final) and the reaction was incubated at 95 °C for 10 min following by 5 min at 4 °C. DNA cleavage was monitored on a 20% denaturing acrylamide gel (8 M urea, 1× TAE buffer) and run at 65 °C for 150 min at 150 V. DNA oligonucleotide probes were synthetized by Thermo Fisher Scientific. The DNA oligonucleotide probe sequence used in this study is:

5′-(6-FAM)-TTTTGCAATTA<u>AC</u>AATTGCAAAA

**Statistics and reproducibility**. All western blots, DNA gels, and microscopy analysis shown in Figs. 1e–g, 2a, 4c, 5d, h, j, and 7a, c, and in Supplementary Figs. 1C, D, F, G, 2C, 3D, E, H, 4A, B, E, 5A, F, 7B, 9E–G, 11A, E, 12A, E, F, and 13B were repeated successfully at least three times and representative images were shown in this manuscript.

**Reporting summary**. Further information on research design is available in the Nature Research Reporting Summary linked to this article.

## Data availability

Databases used in this study were as follows: Stanford_ChipSeq_GM12878_TNFa_NFKB_IgG-rab (https://www.ncbi.nlm.nih.gov/geo/query/acc.cgi?acc=GSM935478), the human genome GRCh37/hg19 (https://www.ncbi.nlm.nih.gov/assembly/GCF_000001405.13/), and GRCh38/hg38 (https://www.ncbi.nlm.nih.gov/assembly/GCF_000001405.26/). The RNA sequencing data and the ChIP-sequencing data generated in this study have been deposited in the NCBI Sequence Read Archive, using the Bioproject Accession: PRJNA684601. The ChIP-seq data was aligned using GRC37 in order to compare directly with previous studies. The RNA-seq was aligned to the current human genome build (GRCh38). Source data are provided with this paper.

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

## Acknowledgements

We thank Andrew Babbin, Nicholas Pannunzio, Feng Qiao, Matthew Weitzman, and Stephanie Yazinski for comments on the manuscript; Reuben Harris for the APOBEC3A/B antibody obtained through the National Institutes of Health (NIH) AIDS Reagent Program, Division of AIDS, NIAID, NIH; and Junjie Chen for the MCF10A IRF3 KO cell line. We also thank Laurent Roux and Geneviève Mottet-Osman for advice on the growth and quantification of Sendai virus. R.B. is supported by a NIH Pathway to Independence Award (CA212154), a NIH MERIT Award (CA252081), the California Breast Cancer Research Program, the Concern foundation, and a University of California Cancer Research Coordinating Committee (CRCC) award grant. M.S. is supported by an NIH Pathway to Independence Award (HL138193) and a grant from the National Institute of Diabetes and Digestive and Kidney Diseases (DK097771). J.M. is supported by an NIH Pathway to Independence Award (CA212290) and MSKCC core grant P30-CA008748. B.L.S. is supported by a U.S. Public Health Service grant AI026765 from the National Institutes of Health. This work was also made possible, in part, through access to the Genomics High Throughput Facility Shared Resource of the Chao Family Comprehensive Cancer Center Support Grant (P30-CA062203). Cartoons in Figs. 2d, e, 3e, and 8, and in Supplementary Figs. 3G and 4F were created with BioRender.com

## Author contributions

S.O., E.B., D.B., P.J., A.S., I.W., A.D., L.M., G.P.T., and R.B. performed all the experiments. S.O., E.B., and M.S. analyzed RNA-sequencing and ChIP-sequencing data. A.D.

and J.M. generated the MCF10A p53/IRF3 KO cell line. B.L.S. supervised experiments requiring SeV infection. R.B. conceived and designed the study, supervised the project, and wrote the paper.

## Competing interests

The authors declare no competing interests.
