## [Peer Review File · Nature Communications]

REVIEWER COMMENTS

Reviewer #1 (Remarks to the Author):

Overview:

This manuscript presents two mechanisms by which APOBEC3A can be induced in both an immortalised mammary epithelial cell line (MCF10A) and a head and neck cancer cell line (BICR6). Identification of the stimuli and pathways responsible for APOBEC3A induction in epithelial cells is important, as APOBEC3A deaminase activity against genomic DNA has been implicated as a major source of mutations in carcinomas. A lot of clear data are presented, the experiments have been carefully controlled and a convincing case is made for: (1) RIG-I/MAVS-dependent induction of APOBEC3A via paracrine (and autocrine?) type 1 interferon-JAK-STAT2 signalling following viral RNA sensing and (2) induction of APOBEC3A by canonical NFkB signalling upon drug-induced replication stress. As acknowledged by the authors, previous publications have demonstrated type 1 IFN-dependent APOBEC3A induction, with one (reference 39 in this m/s) implicating RIG-I upstream, following transcription of cytoplasmic DNA by RNA polymerase III. The novelty of the current manuscript lies in the use of KO cell lines and siRNA to clearly implicate RIG-I/MAVS RNA sensing upstream and STAT2 activation downstream, of IFN in the induction of APOBEC3A, and in the role of replication stress exacerbated by ATR inhibition. This latter mechanism may have particular relevance to the use of ATR inhibitors currently in clinical development for cancer therapy.

Specific comments:

1. While it is good that the mechanisms responsible for A3A upregulation are demonstrated in two different cell lines (the immortalised breast epithelial cell line MCF10A and a head and neck cancer cell line, BICR6), the interchangeable use of the two cell lines throughout the figures is slightly confusing in places. I would recommend labelling each figure part to clearly indicate whether MCF10A or BICR6 cells have been used.

2. Why are p53 KO cells used for the IRF3 KO experiments shown in Figure 2C and E? Is IRF3 KO not viable on p53 WT background? Some explanation should be given, at least in the methods section.

3. Blotting for STAT1 and STAT3 should be included in Supplementary Figure 4 to show that (a) the siRNAs used for STAT1 and STAT3 did indeed reduce their expression and (b) that the STAT2 siRNAs did not have any effect on STAT1 or STAT3 expression.

4. Consistent with their recently-published work, (Jalili et al Nat Comms 2020), the authors show editing of DDOST mRNA acts as a good readout for APOBEC3A induction. Given the focus of the manuscript on potential means by which mutagenic APOBEC3A may be induced in tumours however, it would be useful to understand whether the level of APOBEC3A induced by these stimuli result in any activity against genomic DNA. This could be assessed in several ways, e.g. monitoring DSBs by gamma-H2AX staining or measurement of abasic sites in genomic DNA. Note that the absence of such activity against genomic DNA would in no way invalidate these findings but the authors might wish to speculate that the loss of additional (posttranslational?) APOBEC3A regulation or DNA repair mechanisms might be necessary for APOBEC3A-mediated mutagenesis to occur.

5. Strong APOBEC3A induction is shown with quite high concentrations / lengthy treatments with hydroxyurea and other compounds or conditions (IR) that would be expected to cause DNA damage in addition to replication stress, and the terms 'replication stress' and 'DNA damage' are used quite interchangeably in the manuscript. It would be useful if the authors could be a bit more precise on this point. Is the NFkB activation they observed caused by stalled replication forks / ssDNA or are DNA breaks required? Can the APOBEC3A induction they observe be inhibited by supplementation with nucleosides?

7. Given the strong potentiation of HU and IR-dependent APOBEC3A induction caused by ATR inhibition, it would be interesting to know whether similar effects are observed upon CHK1

inhibition. I would be surprised if not, and it would further increase the interest from a cancer therapy perspective given that CHK1 inhibitors are also in clinical trials.

7. The model presented in Figure 8, together with previous work from the authors suggests that IFN-stimulated A3A upregulation could in turn generate replication stress and ATR activation. Have they checked whether ATRi further increases A3A expression induced following RIG-I/MAVS activation? This would be interesting to see, as one could imagine that A3A-dependent ATR activation would act as a negative feedback loop by which to shut down A3A expression following IFN activation, again helping to explain the episodic nature of A3A expression (and with further possible implications for the use of ATRi in patients).

8. Related to the previous point, what happens if both RIG-I activation and drug-induced replication stress (HU+ATRi) occur simultaneously? Is the effect on A3A induction additive? It would be interesting to see whether the p53 and STAT2 could simultaneously bind and activate the A3A promoter and whether this would result in a greater increase in A3A expression than either pathway alone.

Reviewer #2 (Remarks to the Author):

Oh, Burnique et al. present data examining the regulation of APOBEC3A (A3A) expression in response to various cellular stresses including viral infection, transfection of oligonucleotides, and treatment with DNA damaging agents. The transient nature of A3A mRNA expression is a key aspect of this report. Importantly, the authors provide signaling mechanisms that may explain the transient induction of A3A in tumors. This is a key advance from the study and will be of general interest to cancer biologists and genome integrity researchers. The study is generally well controlled and shows that differences exist in signaling pathways responsible for A3A induction between genotoxic stress vs. viral infection. Notably, NF- κ B was necessary for A3A induction following HU + ATRi. This is an interesting observation. Despite minimal mechanism being provided to explain these descriptive differences, it still is a key advance. I am positive about this study but feel that several concerns as listed below should be addressed prior to publication.

Major concerns

1. They use MCF10A cells (non-tumorigenic breast epithelial) and BICR6 (tumorigenic, keratinocyte-like squamous cell carcinoma) cell lines to examine the inflammatory signaling leading to the expression of A3A under these treatment conditions. Their conclusion is that the transient expression of A3A occurs via different pathways in the setting of treatment with DNA damaging agents when compared viral infection. While this is an interesting observation and the authors characterize the proteins involved in this signaling under both conditions in sufficient depth, it is unclear whether the findings made in this study will hold true for A3A expression in models other than MCF10A/BICR6. Indeed, the authors report significant differences between MCF10A and BICR6 in their paper, which indicates a need to look beyond these lines alone. Further control experiments are needed to support some of the findings.

2. In Fig. 1A and Fig. S1A, different immunostimulatory DNA and RNA molecules are introduced into MCF10A and BICR6 cells respectively, and the expression of A3A mRNA is measured at 16h post-transfection. In this experiment, A3A was induced at 1.7×10^{-3} % at 16h post-transfection in BICR6 cells (Fig. S1A). The authors then draw upon Fig. 1B to suggest that this induction of A3A expression upon 3p-hpRNA treatment is transient by monitoring A3A levels at 40, 64, and 80h post-transfection in BICR6 cells alone. However, even at 80h post-transfection, the level of A3A transcripts is approximately 1.3×10^{-3} %, which is highly elevated in comparison to non-transfected control cells and comparable to the induction in Fig. S1A. Given that these cell line models are used throughout the paper, it is necessary that the authors provide a longer time course post-transfection to demonstrate that the induction of A3A expression is indeed transient in both MCF10A and BICR6 cells and returns to levels similar to non-transfected cells over time.

3. In Fig. 2, the authors determine the proteins involved in the induction of A3A expression upon 3p-hpRNA transfection by systematic KO of candidate factors in MCF10A cells. However, in

contrast to experiments presented in the remainder of the study, the authors use p53 KO MCF10A cells to examine the effect of IRF3 KO on immune induction upon 3p-hpRNA treatment. The reasoning behind the usage of p53 KO cells at this stage of the study is not apparent and needs to be made clear in the description of these data. If there is no strong reason for using p53 KO at this stage, the authors must use MCF10A, p53 wildtype cells to ensure comparability between their KO cells within the study. If they do elect to justify their usage of p53 KO cells, Western blot analysis needs to be presented to confirm the p53 KO in these cells.

4. The authors further examine the factors involved in the induction of A3A by RNAi of STAT1/2/3. While both STAT1 and STAT2 are phosphorylated upon 3p-hpRNA treatment (Fig. S3D), the authors present data suggesting that only STAT2 knockdown impacts on A3A expression (Fig. 3B). However, no indication of the knockdown efficiency of their RNAi experiment is given and it is essential that the authors provide Western blot analysis confirming the knockdown efficiency in this context. In addition, given the known interplay between STAT1 and STAT2, I suggest that the authors comment on the fact that STAT1 seems to be phosphorylated but dispensable for the induction of A3A in their description of these data.

5. It is unclear why the authors elect to use different drug treatments for MCF10A and BICR6 cells respectively in Fig. 4F-H and Fig. S5D-F. It is essential that data is presented from cell lines in a consistent manner.

6. The authors examine the effect of HU/ATRi treatment on the cell cycle in Fig. S7C and argue that inhibition of cell cycle progression and concomitant micronucleus formation underlies the absent IFN response that they observe under these conditions in comparison to IR/ATRi treatment (Fig. 5g,h; S5). While this is indeed a reasonable hypothesis, the analysis presented to support it is incomplete. In order to thoroughly examine the dependence of this differential response on cell cycle progression in these treatment contexts, the authors should compare, in both of their cell line models (MCF10A and BICR6), untreated, HU, IR alone and in combination with ATRi. In addition, it is necessary to show a representative Flow cytometry plot, indicating the gating strategy applied and plots from a representative experiment showing the different treatment conditions.

7. It is unclear how the strong STAT2 phosphorylation upon IR/ATRi treatment in MCF10A cells (Fig. 5H) is not the direct cause of A3A expression as suggested in lines 300-308 and by analysis of STAT2 KO clones in Fig. 5I. The authors need to account for this discrepancy. Possible explanations include impaired P-STAT2 binding to the A3A promoter upon IR/ATRi treatment, which the authors could examine with their ChIP-qPCR assay. Again, in these experiments, my main concern about this report becomes apparent, eg. the difference between MCF10A and BICR6 cells (Fig. S7G). It is clear that these cells behave differently in regards to damaged-induced, IFN-mediated A3A expression.

8. This is also apparent in their subsequent analysis of the inflammatory gene expression programs by RT-qPCR and RNAseq presented in Fig. 6 and S8. There are inconsistencies between MCF10A and BICR6 in terms of data presentation, which need to be accounted for to show that the findings are consistent between these cell line models. For example, how do the authors justify the selection of different inflammatory genes in Fig. 6C and S8B? It is imperative that this analysis is performed in both MCF10A and BICR6 cell lines to validate their findings, perhaps presentation of such data in a manner analogous Fig. 6B to illustrate any common changes and potential differences between their models.

Minor comments:

1. In lines 140-143, the authors analyze STING KO in order to “rule out potential interconnection between RIG-I/MAVS [and] STING”. They state that STING KO “did not impact A3A after 3p-hpRNA transfection (Figure 1D)”. This is not the case. Both STING KO clones show increased A3A expression upon 3p-hpRNA transfection, especially clone #8, where there is nearly two-fold increased A3A expression when compared to non-transfected control (from approximately 0.8 to 1.4×10^{-3} %). Consistently, when examining the impact of STING KO on the induction of A3A, the same STING KO clones, which the authors examine in this context show the same two-fold increased A3A expression compared to wildtype controls upon HU treatment (Fig.5B). The authors

need to comment on these differences in their description of the data.

2. In both Fig. 7B, S9B,C as well as Fig. S6C, S3A, the authors quantify recruitment of transcription factors p65 and IRF3 to the nucleus by immunofluorescence respectively. Yet, the data is presented differently. I recommend that the authors are consistent in their quantification and presentation of these data.

3. Please also consider the following text corrections:

Line 55: "response, mechanism by which viral infection triggers A3A expression are still poorly understood."

Line 71: "Using a yeast model, Gordenin and colleagues [...]"

Line 130: "expression still remains to be demonstrated"

Line 180: "We next asked whether direct RIG-I stimulation [...]"

Line 259: "[...] the absence of RIG-I did not affect the induction of A3A mRNA [...]"

Fig. S8C: The label of this panel should read "KO STAT2".

Reviewer #3 (Remarks to the Author):

In this manuscript Rémi Buisson's group trying to understand the mechanisms that control APOBEC3A (A3A) expression during viral infection and DNA damage. In the first part of the manuscript, they showed that viral PAMPs induce A3A expression in RIG-I/MDA5-MAVS dependent manner. In the second part of the manuscript, they show that DNA damage-induced expression of A3A is dependent on p65/NFKB pathway. The experiments performed are clear and convincing. However, there is an almost negligible novelty in the work and does not increase our knowledge in the field.

Major Points

APOBEC's is an established ISG and more than 100 papers might have shown that APOBEC's are induced by RIG-I/MDA5-MAVS dependent IFN response. This is also very well known in the field that APOBEC's (actually most ISG's) are regulated by JAK/STAT1/2 signaling especially during viral infection. There are more than 200 ISG's and each one can be used to perform such analysis resulting in a manuscript. Their own transcriptomic data suggest that several of the ISG's (including A3A) are regulated in a similar fashion. As a whole, this part is providing no new knowledge to the field.

It was interesting to note that DNA damage-induced A3A expression is not dependent on RIG-MAVS signaling or JAK/STAT signaling in cancer cells and p65/NFKB play important role in the regulation of A3A during DNA damage in cancer cells. Again, it has been shown previously in papers published in Cancer Research and BBRC journals that p65 or RELB binds in A3B promoter and important for transcription of A3B in cancer cells (PMID: 27577680; PMID: 26420215). So it's not surprising that A3A is also controlled by NFKB where it is very well known that DDR induces NFKB signaling (PMID: 28626800). There is some new information here (A3A vs A3B) but I doubt that this is enough to get in this journal.

In nutshell, there is no doubt that the study is performed very nicely. However, the knowledge that comes from this manuscript is incremental.

Minor Points

1. Why author did not check the role of TLR3 or IF16 in the regulation of APOBEC3A?
2. Both the introduction and discussion have too much information and very extensive review of the literature, several of the things are not very relevant to the manuscript. This can be reduced.
3. This is surprising that DNA damage-induced expression of A3A is not dependent on IFN response. Previously, it has been shown that DNA damage induces IFN response via cGAS-STING pathways (PMID: 28738408 PMID: 22013119 PMID: 25692705). More specifically, a manuscript published in the NAR journal (PMID: 28100701) showed that "cytoplasmic DNA triggers interferon α and β production through the RNA polymerase III transcription/RIG-I pathway leading to massive upregulation of APOBEC3A". Is it possible that the model used in your manuscript to cause DNA Damage (HU+ATRi) is different from previous works? This discrepancy should be resolved.

REVIEWER COMMENTS

Reviewer #1 (Remarks to the Author):

Overview:

This manuscript presents two mechanisms by which APOBEC3A can be induced in both an immortalised mammary epithelial cell line (MCF10A) and a head and neck cancer cell line (BICR6). Identification of the stimuli and pathways responsible for APOBEC3A induction in epithelial cells is important, as APOBEC3A deaminase activity against genomic DNA has been implicated as a major source of mutations in carcinomas. A lot of clear data are presented, the experiments have been carefully controlled and a convincing case is made for: (1) RIG-I/MAVS-dependent induction of APOBEC3A via paracrine (and autocrine?) type 1 interferon-JAK-STAT2 signalling following viral RNA sensing and (2) induction of APOBEC3A by canonical NFkB signalling upon drug-induced replication stress. As acknowledged by the authors, previous publications have demonstrated type 1 IFN-dependent APOBEC3A induction, with one (reference 39 in this m/s) implicating RIG-I upstream, following transcription of cytoplasmic DNA by RNA polymerase III. The novelty of the current manuscript lies in the use of KO cell lines and siRNA to clearly implicate RIG-I/MAVS RNA sensing upstream and STAT2 activation downstream, of IFN in the induction of APOBEC3A, and in the role of replication stress exacerbated by ATR inhibition. This latter mechanism may have particular relevance to the use of ATR inhibitors currently in clinical development for cancer therapy.

We thank the reviewer for his/her appreciation of the significance and quality of our work. We have now addressed all the reviewer comments.

Specific comments:

1. While it is good that the mechanisms responsible for A3A upregulation are demonstrated in two different cell lines (the immortalised breast epithelial cell line MCF10A and a head and neck cancer cell line, BICR6), the interchangeable use of the two cell lines throughout the figures is slightly confusing in places. I would recommend labelling each figure part to clearly indicate whether MCF10A or BICR6 cells have been used.

In this manuscript, it was important for us to confirm all the key data in both MCF10A and BICR6 cells to ensure reproducibility of the results across cell lines. We have now labeled the figures to clearly indicate whether MCF10A or BICR6 was used. Thanks for this suggestion.

2. Why are p53 KO cells used for the IRF3 KO experiments shown in Figure 2C and E? Is IRF3 KO not viable on p53 WT background? Some explanation should be given, at least in the methods section.

In these panels, we used IRF3 KO cells in a MCF10A p53 KO background because these cell lines were directly available to us. We originally generated these cell lines for another project. However, we found that the absence of p53 did not affect A3A expression compared to WT cells after 3p-hpRNA transfection, thus MCF10A p53 KO could still be a good model to determine the role of IRF3 in the regulation of A3A expression. We have now added these important control experiments (**new Supplementary Figures 3E, 3F and 3G**). In addition, to further support our conclusion, we have now repeated all of these experiments in MCF10A WT background

knockout for IRF3 (provided to us by Junjie Chen's lab at MD Anderson) and obtained the same results (**new Figures 2C, 2E, and Supplementary Figure 3D**). Together, these new results strongly support our conclusion that IRF3 is essential to promote A3A expression after viral infection.

3. Blotting for STAT1 and STAT3 should be included in Supplementary Figure 4 to show that (a) the siRNAs used for STAT1 and STAT3 did indeed reduce their expression and (b) that the STAT2 siRNAs did not have any effect on STAT1 or STAT3 expression.

We would like to apologize for omitting the western blot panels confirming the knockdown efficiency of siRNA against STAT1 and STAT3. We now display these essential control experiments (**new Supplementary Figures 4A and 4B**). In addition, as the reviewer suggested, we now show that siSTAT2 has no effect on the level of STAT1 or STAT3 in both MCF10A and BICR6 cell lines (**new Supplementary Figure 4A**).

4. Consistent with their recently-published work, (Jalili et al Nat Comms 2020), the authors show editing of DDOST mRNA acts as a good readout for APOBEC3A induction. Given the focus of the manuscript on potential means by which mutagenic APOBEC3A may be induced in tumours however, it would be useful to understand whether the level of APOBEC3A induced by these stimuli result in any activity against genomic DNA. This could be assessed in several ways, e.g. monitoring DSBs by gamma-H2AX staining or measurement of abasic sites in genomic DNA. Note that the absence of such activity against genomic DNA would in no way invalidate these findings but the authors might wish to speculate that the loss of additional (posttranslational?) APOBEC3A regulation or DNA repair mechanisms might be necessary for APOBEC3A-mediated mutagenesis to occur.

This is a very good point! To demonstrate that endogenous expression of A3A has potential activity against DNA and not only RNA-editing activity, we have now added several new experiments. First, we performed a DNA deaminase activity assay using a hairpin DNA substrate that we recently described as a specific target of A3A and can be used as a readout of A3A deaminase activity on DNA (Langenbucher et al. Nature Communications, 2021). Cell extracts expressing endogenous A3A induced by 3p-hpRNA or HU+ATRi treatment showed a strong A3A DNA deaminase activity compared to cell extracts made from untreated cells (**new Figure 1E and new supplementary Figure 5F**). Second, we previously published that A3A expression from a plasmid-based system increases the level of DNA replication stress leading to the activation of the ATR-CHK1 pathway (Buisson et al. Cancer Research, 2017). To determine whether endogenous expression of A3A also increases the level of replication stress and thus directly targets genomic DNA, we transfected cells with 3p-hpRNA and monitored the level of CHK1 phosphorylation (**new Figures 1F-G**). Consistent with our previous work, cells expressing A3A present a higher level of CHK1 phosphorylation, and knocking down of A3A suppressed CHK1 activation. Together, these new results suggest that A3A induction directly results in activity against genomic DNA.

5. Strong APOBEC3A induction is shown with quite high concentrations / lengthy treatments with hydroxyurea and other compounds or conditions (IR) that would be expected to cause DNA damage in addition to replication stress, and the terms 'replication stress' and 'DNA damage' are used quite interchangeably in the manuscript. It would be useful if the authors could be a bit more precise on this point. Is the NFkB activation they observed caused by stalled replication

forks / ssDNA or are DNA breaks required? Can the APOBE3A induction they observe be inhibited by supplementation with nucleosides?

The reviewer raises another very good point. Prolonged HU treatment or other drug-induced replication stress leads to the collapse of stalled replication forks and the formation of DNA double-strand breaks. We now monitored over time CHK1 phosphorylation (as a DNA replication stress marker) and H2AX phosphorylation (as collapse replication fork and DNA double-strand break markers) following HU treatment. While CHK1 phosphorylation is quickly induced by HU, H2AX phosphorylation only occurs 24-32h after HU treatment at a similar time as A3A expression, suggesting that A3A expression and fork collapse are concomitant (**new Supplementary Figure 5A compared to Figure 4A and Supplementary Figure 5B**). Moreover, we now show that the nuclear localization of p65 occurs only in gH2AX positive cells (**new Supplementary Figure 11D**) implying that the collapse of the forks is important for the activation of the NFKB pathway and so A3A expression. Together, these new results suggest that the fork collapse is an important step to promote A3A expression after DNA replication stress. To avoid further confusion, we have now replaced “replication stress” with “replication stress-induced DNA damage”.

As mentioned by the reviewer, cells supplemented with nucleosides have been shown to prevent replication stress and replication stress-associated phenotypes. Following a published protocol from Kanu et al. Genome Biology 2016, we now show A3A expression is reduced after HU or HU+ATRi treatment in cells supplemented with nucleosides (**Rebuttal Figure 1**). If the reviewer deems it necessary, we will be happy to include these new panels in the revised manuscript.

Rebuttal Figure 1. MCF10A or BICR6 cells were pre-incubated in media supplemented with 300uM of nucleoside (EmbryoMax Nucleosides, Sigma) for 24h. The cells were then treated with HU or HU+ATRi for 32h in media supplemented with fresh nucleoside. A3A expression level was determined by RT-qPCR. Error bar: S.D. (n = 3).

7. Given the strong potentiation of HU and IR-dependent APOBEC3A induction caused by ATR inhibition, it would be interesting to know whether similar effects are observed upon CHK1 inhibition. I would be surprised if not, and it would further increase the interest from a cancer therapy perspective given that CHK1 inhibitors are also in clinical trials.

To determine whether ATR prevents A3A expression after replication stress-induced DNA damage through the ATR-CHK1 signaling axis, we treated cells with CHK1 inhibitors (CCT244747) and HU. Similar to ATRi, CHK1i further enhance A3A expression in both MCF10A and BICR6 cells treated with HU (**new Figure 4H and new Supplementary Figure 6D**) suggesting that ATR-CHK1 signaling axis is important to prevent replication stress-induced A3A expression. We thank the reviewer for this great suggestion!

7. The model presented in Figure 8, together with previous work from the authors suggests that IFN-stimulated A3A upregulation could in turn generate replication stress and ATR activation.

Have they checked whether ATRi further increases A3A expression induced following RIG-I/MAVS activation? This would be interesting to see, as one could imagine that A3A-dependent ATR activation would act as a negative feedback loop by which to shut down A3A expression following IFN activation, again helping to explain the episodic nature of A3A expression (and with further possible implications for the use of ATRi in patients).

This is a great question! To determine whether ATR activity is important to prevent A3A expression following RIG-I / MAVS activation, we treated cells with both ATRi and 3p-hpRNA (**new Supplementary Figures 5E**). ATR inhibition did not affect A3A expression compared to 3p-hpRNA alone suggesting that ATR is not involved in the interferon-induced A3A expression. Together, these new results further confirm our main conclusion that the two pathways regulating A3A expression after viral infection and DNA damage are independent of each other.

8. Related to the previous point, what happens if both RIG-I activation and drug-induced replication stress (HU+ATRi) occur simultaneously? Is the effect on A3A induction additive? It would be interesting to see whether the p65 and STAT2 could simultaneously bind and activate the A3A promoter and whether this would result in a greater increase in A3A expression than either pathway alone.

This is another great question! To determine whether A3A induction is additive when both RIG-I activation and drug-induced replication stress occur simultaneously, we treated cells with HU+ATRi, 3p-hpRNA, or both (**new Supplementary Figures 7H**). While cells treated with HU+ATRi or 3p-hpRNA induce a similar level of A3A, cells treated with both HU+ATRi and 3p-hpRNA express a higher level of A3A. This increase of A3A level is additive with about twice as much expression after both treatments suggesting that both pathways can be activated at the same time to promote A3A expression. Moreover, the fact that A3A induction is additive implies that the two pathways regulating A3A expression are independent of each other, further confirming our main conclusion.

Reviewer #2 (Remarks to the Author):

Oh, Bournique et al. present data examining the regulation of APOBEC3A (A3A) expression in response to various cellular stresses including viral infection, transfection of oligonucleotides, and treatment with DNA damaging agents. The transient nature of A3A mRNA expression is a key aspect of this report. Importantly, the authors provide signaling mechanisms that may explain the transient induction of A3A in tumors. This is a key advance from the study and will be of general interest to cancer biologists and genome integrity researchers. The study is generally well controlled and shows that differences exist in signaling pathways responsible for A3A induction between genotoxic stress vs. viral infection. Notably, NF- κ B was necessary for A3A induction following HU + ATRi. This is an interesting observation. Despite minimal mechanism being provided to explain these descriptive differences, it still is a key advance. I am positive about this study but feel that several concerns as listed below should be addressed prior to publication.

We thank the reviewer for his/her appreciation of our work and for the excellent suggestions to strengthen it.

Major concerns:

1. They use MCF10A cells (non-tumorigenic breast epithelial) and BICR6 (tumorigenic, keratinocyte-like squamous cell carcinoma) cell lines to examine the inflammatory signaling leading to the expression of A3A under these treatment conditions. Their conclusion is that the transient expression of A3A occurs via different pathways in the setting of treatment with DNA damaging agents when compared viral infection. While this is an interesting observation and the authors characterize the proteins involved in this signaling under both conditions in sufficient depth, it is unclear whether the findings made in this study will hold true for A3A expression in models other than MCF10A/BICR6. Indeed, the authors report significant differences between MCF10A and BICR6 in their paper, which indicates a need to look beyond these lines alone. Further control experiments are needed to support some of the findings.

This is a very important point and we agree that it is crucial to validate our findings in different cell lines to ensure reproducibility of the results. Across our manuscript, we validated all the key results with both MCF10A and BICR6 cells because it was important for us to confirm that our findings were not cell line specific (Please see reply to comment 7 about the differences between MCF10A and BICR6). We have now increased our panel of cell lines and confirmed the key findings. We selected PC-9, TPH-1, and RPE-1 cell lines, three model cell lines that have been previously used to study A3A (Land et al. JBC 2013 / Buisson et al. Cancer Research, 2017 / Green et al. Cancer Research, 2017 / Jalili et al. Nature Communications, 2020). First, we confirmed the induction of A3A after 3p-hpRNA transfection or HU+ATRi treatment in these new cell lines (**new Supplementary Figures 4D and 12I**). We then knocked down STAT2 or p65 followed by 3p-hpRNA transfection or HU+ATRi treatment respectively. Similar to the results obtained in BICR6 and MCF10A cells, the knockdown of STAT2 or p65 decreases A3A expression (**new Supplementary Figures 4D and 12I**). Together, these new data strongly support our proposed A3A regulation mechanisms in models beyond MCF10A and BICR6 cells.

2. In Fig. 1A and Fig. S1A, different immunostimulatory DNA and RNA molecules are introduced into MCF10A and BICR6 cells respectively, and the expression of A3A mRNA is measured at 16h post-transfection. In this experiment, A3A was induced at 1.7×10^{-3} % at 16h post-transfection in BICR6 cells (Fig. S1A). The authors then draw upon Fig. 1B to suggest that this induction of A3A expression upon 3p-hpRNA treatment is transient by monitoring A3A levels at 40, 64, and 80h post-transfection in BICR6 cells alone. However, even at 80h post-transfection, the level of A3A transcripts is approximately 1.3×10^{-3} %, which is highly elevated in comparison to non-transfected control cells and comparable to the induction in Fig. S1A. Given that these cell line models are used throughout the paper, it is necessary that the authors provide a longer time course post-transfection to demonstrate that the induction of A3A expression is indeed transient in both MCF10A and BICR6 cells and returns to levels similar to non-transfected cells over time.

The recovery of A3A levels after 3p-hpRNA transfected cells depends on how quickly the cells can clear the 3p-hpRNA from their cytoplasm. As suggested by the reviewer, we have now repeated the time course in both MCF10A and BICR6 cells and increased the recovery time up to 12 days post-transfection (**new Figure 1B and new Supplementary Figure 1B**). We now can monitor over time A3A expression levels up to its return to basal levels, similar to non-transfected cells in both model cell lines. Thanks for pointing this out!

3. In Fig. 2, the authors determine the proteins involved in the induction of A3A expression upon 3p-hpRNA transfection by systematic KO of candidate factors in MCF10A cells. However, in contrast to experiments presented in the remainder of the study, the authors use p53 KO MCF10A cells to examine the effect of IRF3 KO on immune induction upon 3p-hpRNA treatment. The reasoning behind the usage of p53 KO cells at this stage of the study is not apparent and needs to be made clear in the description of these data. If there is no strong reason for using p53 KO at this stage, the authors must use MCF10A, p53 wildtype cells to ensure comparability between their KOs within the study. If they do elect to justify their usage of p53 KO cells, Western blot analysis needs to be presented to confirm the p53 KO in these cells.

In these panels, we used IRF3 KO cells in a MCF10A p53 KO background because these cell lines were directly available to us. We originally generated these cell lines for another project. However, we found that the absence of p53 did not affect A3A expression compared to WT cells after 3p-hpRNA transfection; thus, MCF10A p53 KO could still be a good model to determine the role of IRF3 in the regulation of A3A expression. We have now added these important control experiments as suggested by the reviewer (**new Supplementary Figures 3E, 3F and 3G**). In addition, to further support our conclusion, we have now repeated all of these experiments in MCF10A WT background knockout for IRF3 (provided to us by Junjie Chen's lab at MD Anderson) and obtained the same results (**new Figure 2C, 2E, and Supplementary Figure 3D**). Together, these new results strongly support that IRF3 is essential to promote A3A expression after viral infection.

4. The authors further examine the factors involved in the induction of A3A by RNAi of STAT1/2/3. While both STAT1 and STAT2 are phosphorylated upon 3p-hpRNA treatment (Fig. S3D), the authors present data suggesting that only STAT2 knockdown impacts on A3A expression (Fig. 3B). However, no indication of the knockdown efficiency of their RNAi experiment is given and it is essential that the authors provide Western blot analysis confirming the knockdown efficiency in this context. In addition, given the known interplay between STAT1 and STAT2, I suggest that the authors comment on the fact that STAT1 seems to be phosphorylated but dispensable for the induction of A3A in their description of these data.

We apologize for omitting the western blot panels confirming the knockdown efficiency of siRNA against STAT1 and STAT3. We have now added these essential control experiments (**new Supplementary Figures 4A and 4B**). In addition, we now show that siRNA against STAT2 does not affect either STAT1 or STAT3 levels (**new Supplementary Figure 4A**). The reviewer raises a good point that STAT1 is also phosphorylated after 3p-hpRNA transfection but STAT1 knockdown did not affect A3A level. It is well established that the activation of the JAK pathway leads to the upregulation of many Interferon Stimulated Genes (ISGs) through the phosphorylation of different STAT transcription factors (O'Shea et al. Annu. Rev. Med, 2015). However, the expression levels of the ISGs are not necessarily controlled by the same STATs even if several STATs are phosphorylated at the same time. STAT2 has been shown to regulate ISG expression alone or in complex with STAT1, explaining why the knockdown of STAT1 did not affect A3A levels after 3p-hpRNA transfection since STAT2 can act by itself. However, we cannot exclude that the STAT1-STAT2 complex has no function regulating A3A levels. Still, in the absence of STAT1, STAT2 alone is sufficient to compensate for the loss of STAT1 and induce A3A expression. We have now added a comment on this specific point in our revised manuscript: **"While STAT2 regulates many ISGs in complex with STAT1, STAT2 is also known to regulate genes by itself explaining why the knockdown of STAT1 did not affect A3A levels after 3p-hpRNA transfection. However, we cannot exclude the possibility that the STAT1-STAT2 complex has no function regulating A3A**

levels. Regardless, in the absence of STAT1, STAT2 alone is sufficient to compensate for the loss of STAT1 and induce A3A expression”

5. It is unclear why the authors elect to use different drug treatments for MCF10A and BICR6 cells respectively in Fig. 4F-H and Fig. S5D-F. It is essential that data is presented from cell lines in a consistent manner.

We apologize for this inconsistency between drug treatments across cell lines. We have now revised these panels and show the same drug treatments for both MCF10A and BICR6 cells in order to have consistent data in this section of the manuscript (**new Figures 4E, 4F, 4G, and 4H compared to new Supplementary Figures 6A, 6B, 6C, and 6D** previous Fig. 4F-H and Fig. S5D-F).

6. The authors examine the effect of HU/ATRi treatment on the cell cycle in Fig. S7C and argue that inhibition of cell cycle progression and concomitant micronucleus formation underlies the absent IFN response that they observe under these conditions in comparison to IR/ATRi treatment (Fig. 5g,h; S5). While this is indeed a reasonable hypothesis, the analysis presented to support it is incomplete. In order to thoroughly examine the dependence of this differential response on cell cycle progression in these treatment contexts, the authors should compare, in both of their cell line models (MCF10A and BICR6), untreated, HU, IR alone and in combination with ATRi. In addition, it is necessary to show a representative Flow cytometry plot, indicating the gating strategy applied and plots from a representative experiment showing the different treatment conditions.

Following reviewer suggestions, we repeated our cell cycle analysis and micronuclei quantification in both MCF10A and BICR6 cells, but this time including samples treated with HU or ATRi alone in addition to HU+ATRi. We also show representative flow cytometry plots with the gating strategy we used to quantify the different phases of the cell cycles. Moreover, we now perform cell cycle analysis and micronuclei quantification of irradiated cells treated with or without ATRi in both MCF10A and BICR6. (**new Supplementary Figure 8**)

We now show, as previously reported, that irradiated cells treated with ATRi are still partially cycling and accumulate a high level of micronuclei. The percentage of cycling cell and micronuclei detected after irradiation is similar to these studies (Harding et al. Nature, 2017 / Feng et al. EMBO J, 2020 / Chen et al. Cell Reports, 2020) (**new Supplementary Figures 8A and 8C**). However, cells treated with HU or HU+ATRi are depleted of cycling cells, illustrated by the absence of EdU positive cells and micronuclei (**new Supplementary Figures 8B and 8D**). Together these new results strengthen our previous observation and previous published studies that only cells treated with irradiation are still partially cycling, explaining why these cells accumulate a high level of micronucleus through missegregation of chromosomes during mitosis while after HU, cells are blocked in S-phase.

7. It is unclear how the strong STAT2 phosphorylation upon IR/ATRi treatment in MCF10A cells (Fig. 5H) is not the direct cause of A3A expression as suggested in lines 300-308 and by analysis of STAT2 KO clones in Fig. 5I. The authors need to account for this discrepancy. Possible explanations include impaired P-STAT2 binding to the A3A promoter upon IR/ATRi treatment, which the authors could examine with their ChIP-qPCR assay. Again, in these experiments, my main concern about this report becomes apparent, eg. the difference between

MCF10A and BICR6 cells (Fig. S7G). It is clear that these cells behave differently in regards to damaged-induced, IFN-mediated A3A expression.

This is an important point! We now show that STAT2 phosphorylation in MCF10A cells after IR+ATRi treatment is very modest compared to STAT2 phosphorylation in cells transfected with 3p-hpRNA (**new Figure 5J**). This difference in the level of STAT2 phosphorylation correlates with the level of IFN β following the same treatments (**Supplementary Figure 9C**), implying that IR+ATRi does not stimulate the IFN response as much as 3p-hpRNA transfection. On the other hand, A3A expression is induced at a similar level after 3p-hpRNA or IR+ATRi (**Figure 5K**) suggesting that the induction of A3A requires an additional pathway that we later show to be the canonical NFKB pathway. However, a small fraction of A3A expression is likely the result of the activation of STAT2 after IR+ATRi. As suggested by the reviewer, we monitored the recruitment of STAT2 by ChIP-qPCR on the A3A promoter after IR+ATRi compared to 3p-hpRNA. Similar to the STAT2 phosphorylation level, the binding of STAT2 to the A3A promoter is very modest after IR+ATRi compared to STAT2 recruitment after 3p-hpRNA transfection (**new Supplementary Figure 9D**). This modest recruitment of STAT2 on the A3A promoter may lead to some expression that could explain the small decrease of A3A expression in the absence of STAT2 after IR+ATRi (**Figure 5I**), but overall the level of A3A expression after IR+ATRi is independent of STAT2 activity.

As mentioned by the reviewer, MCF10A and BICR6 cells respond differently to irradiation treatment. While irradiation induces an IFN response in MCF10A cells, we did not detect any activation of the IFN response in BICR6 cells after IR or IR+ATRi (**Supplementary Figure 9E-F**). Previous reports demonstrated that deficiency in non-homologous end-joining abrogates micronuclei formation to prevent cGAS-STING-dependent IFN signaling in response to IR-induced DNA damage (Harding et al. Nature, 2017 / Chen et al. Cell Reports, 2020). Alternatively, many cancer cell lines downregulate important factors such as cGAS or STING to counteract the detection of micronuclei to escape the immune surveillance system (Xia et al. Cell Reports, 2016 / Konno et al. Oncogene, 2018 / Chen et al. Cell Reports, 2020). Here, our results show that micronuclei are still induced after IR in BICR6 cells, suggesting that the absence of IFN response is not the result of a difference in cell cycle progression and micronuclei formation (**new Supplementary Figure 8A and 8B**). However, STING is not (or almost not) expressed in BICR6 cells (**new Supplementary Figure 9G**) similar to many other cancer cell lines that have been reported with a lack of STING expression (Xia et al. Cell Reports, 2016). This result suggests that after irradiation, micronuclei are not properly detected and could explain the defect of IFN response in this cell line. However, BICR6 cells still strongly upregulate A3A mRNA level after IR or IR+ATRi (**Figures 5I**) further supporting that genotoxic stress induces A3A expression through an IFN-independent signaling pathway. We have now updated our manuscript accordingly to mention and discuss this important point.

8. This is also apparent in their subsequent analysis of the inflammatory gene expression programs by RT-qPCR and RNAseq presented in Fig. 6 and S8. There are inconsistencies between MCF10A and BICR6 in terms of data presentation, which need to be accounted for to show that the findings are consistent between these cell line models. For example, how do the authors justify the selection of different inflammatory genes in Fig. 6C and S8B? It is imperative that this analysis is performed in both MCF10A and BICR6 cell lines to validate their findings, perhaps presentation of such data in a manner analogous Fig. 6B to illustrate any common changes and potential differences between their models.

The reviewer raises another good point. We have now revised our data presentation and show the exact same inflammatory genes for both MCF10A and BICR6 cell lines (**Figure 6C and new supplementary Figure 10B**). As expected, when comparing cell lines, a few genes are differentially expressed between MCF10A and BICR6 cells, and we previously only focused on the genes regulated in MCF10A in the same way as in BICR6 cells. As suggested by the reviewer, we have now added a new panel that illustrates the differences between the two cell lines with respect to genes that are regulated similarly versus the few genes that are not (**new supplementary Figure 10C**).

Minor comments:

1. In lines 140-143, the authors analyze STING KO in order to “rule out potential interconnection between RIG-I/MAVS [and] STING”. They state that STING KO “did not impact A3A after 3p-hpRNA transfection (Figure 1D)”. This is not the case. Both STING KO clones show increased A3A expression upon 3p-hpRNA transfection, especially clone #8, where there is nearly two-fold increased A3A expression when compared to non-transfected control (from approximately 0.8 to 1.4×10^{-3} %). Consistently, when examining the impact of STING KO on the induction of A3A, the same STING KO clones, which the authors examine in this context show the same two-fold increased A3A expression compared to wildtype controls upon HU treatment (Fig.5B). The authors need to comment on these differences in their description of the data.

We agree with the reviewer that in the absence of STING, we repeatedly detected an increase of A3A level after 3p-hpRNA transfection or HU treatment. One potential explanation for this increase is that the RIG-I/MDA5/MAVS pathway gets hyperactivated in order to partially compensate for the inactivation of the STING pathway. This increase of A3A expression is observed regardless of the treatment type and also at the basal level in the untreated cells. We have now modified our statements in our manuscript to account for these differences. “**The absence of STING did not suppress A3A expression after 3p-hpRNA transfection (Supp Fig 1H (previous Fig, 1D)), ...**” and “**The slight increase of A3A expression in STING knockout cell lines may be the result of a hyperactivation of the RIG-I / MAVS pathway to compensate for the absence of STING in these cell lines.**”

2. In both Fig. 7B, S9B,C as well as Fig. S6C, S3A, the authors quantify recruitment of transcription factors p65 and IRF3 to the nucleus by immunofluorescence respectively. Yet, the data is presented differently. I recommend that the authors are consistent in their quantification and presentation of these data.

We have now modified our quantification analysis across these panels for consistency in the presentation of these data (**new Supplementary Figure 3A, 3B, and 7C**). In addition, we now show the quantification for both MCF10A and BICR6 cells rather than only one cell line.

3. Please also consider the following text corrections:

Line 55: “response, mechanism by which viral infection triggers A3A expression are still poorly understood.”

Line 71: “Using a yeast model, Gordenin and colleagues [...]”

Line 130: “expression still remains to be demonstrated”

Line 180: “We next asked whether direct RIG-I stimulation [...]”

Line 259: “[...] the absence of RIG-I did not affect the induction of A3A mRNA [...]”

Fig. S8C: The label of this panel should read “KO STAT2”.

We have now made all these text corrections. We thank the reviewer for pointing them out!

Reviewer #3 (Remarks to the Author):

In this manuscript Rémi Buisson's group trying to understand the mechanisms that control APOBEC3A (A3A) expression during viral infection and DNA damage. In the first part of the manuscript, they showed that viral PAMPs induce A3A expression in RIG-I/MDA5-MAVS dependent manner. In the second part of the manuscript, they show that DNA damage-induced expression of A3A is dependent on p65/NFKB pathway. The experiments performed are clear and convincing. However, there is an almost negligible novelty in the work and does not increase our knowledge in the field.

We thank the reviewer for his/her strong appreciation of the quality of our work. We would like to take the opportunity to better explain (below) the novelty and new concepts of our study that we may have missed highlighting properly in order to convey that our work significantly increases our knowledge in the field.

Major Points

APOBEC's is an established ISG and more than 100 papers might have shown that APOBEC's are induced by RIG-I/MDA5-MAVS dependent IFN response. This is also very well known in the field that APOBEC's (actually most ISG's) are regulated by JAK/STAT1/2 signaling especially during viral infection. There are more than 200 ISG's and each one can be used to perform such analysis resulting in a manuscript. Their own transcriptomic data suggest that several of the ISG's (including A3A) are regulated in a similar fashion. As a whole, this part is providing no new knowledge to the field.

We agree with the reviewer that APOBEC3 family members including A3A and A3B are known ISGs and we referenced in our manuscript the publications related to this previous work. However, even though all APOBEC3 proteins are ISGs, they are not all expressed following the same stress, and it is still not well understood which specific stresses stimulate which specific APOBEC3s. To illustrate this differential regulation of the APOBEC3 family, we now compare A3A expression to all the other APOBEC3 members. We demonstrate that after viral infection, A3A is highly induced but not A3B, A3C, A3D, or A3H and we only detect a modest increase of A3F and A3G expression (**new Supplementary Figure 2G**). Together these results strongly suggest that APOBEC3 family members' regulation differs regarding the type of stress. Importantly, not all the APOBEC3 members are induced by viral infection. Thus, the mechanism described in our study is particularly relevant for A3A regulation (as the most highly regulated in this manner). Moreover, this observation confirms the importance of studying individual APOBEC3 members and not necessarily assuming that all the APOBEC3s are regulated in a similar way. APOBEC3 members have been shown to target different substrates and so different APOBEC3s likely respond to different types of stimulus or attack that cells may encounter. It may be unsafe or inefficient for the cells to express all APOBEC3s at the same time. Here, we reveal the detailed mechanism of how viral infection-induced IFN-response leads to A3A expression. Further studies will be necessary to understand better how the other APOBEC3s are regulated in cells.

From the over 200 known ISGs, A3A is particularly important to study with direct clinical implications. A3A and A3B have recently emerged from cancer genomics studies as key drivers of mutations in cancers. In fact, an APOBEC-mutational signature is the second-most common endogenous mutation source across cancer just after aging (Alexandrov et al. Nature, 2020 / Petljak et al. Cell, 2019), and we recently found that A3A is responsible for most of the APOBEC-mutational signatures detected in patient tumors (Buisson et al. Science, 2019 / Jalili

et al. Nature Communications, 2020 / Petljak et al. BioRxiv). To further support that A3A induction results in activity against genomic DNA thus directly impacting the cells, we now monitor A3A DNA deaminase activity (**new Figure 1E**) and show that A3A activity induced by IFN-signaling response increases DNA replication stress (**new Figures 1F and 1G** and see reply to comment 4 of reviewer 1). Determining precisely how cancer cells regulate A3A expression is crucial for the future of targeted therapy in order to suppress A3A expression and to prevent mutations leading to cancer progression, drug resistance, and metastasis.

To determine how cancer cells up-regulate A3A, the main goal of our study was 1) to identify uncharacterized key players essential for A3A expression during the IFN response, 2) generate genetically engineered cell lines deficient for A3A induction by the IFN response, and then 3) use these cell lines to identify new types of stress stimulating A3A expression independently of the IFN response. In this study, for the first time, we have demonstrated that there are two fully independent pathways that modulate A3A expression in cells. However, without an initial pinpoint characterization on how the IFN response regulates A3A expression, we would not have been able to properly and convincingly separate these two pathways. We then used RIG-I and STAT2 KO cell lines as a “tool” to abrogate IFN response-induced A3A and to show that replication stress and DNA damage stimulate A3A expression through a different pathway. Together, our results establish a new concept in the field where A3A expression is controlled through both IFN-dependent and independent mechanisms that, to our knowledge, has never been proposed before. The ability for cancer cells to increase the level of A3A through independent routes begins to explain why the A3A-mutational signature is one of the most common signatures detected in tumors with many different types of stress leading to an upregulation of A3A.

To date and to our knowledge, it has never been demonstrated that A3A or any other APOBEC3 members are regulated by an IFN response that is dependent on RIG-I/MDA5-MAVS. A previous paper elegantly reported a strong up-regulation of A3A by mitochondrial DNA as well as a correlation between A3A expression and both the activation of RIG-I and STING pathway (Suspène et al. NAR, 2017). However, this report did not demonstrate that either RIG-I or STING are required for the induction of A3A by mitochondrial DNA but instead focused on the importance of RNA polymerase III in this process. Here, we performed genetic analysis of the pathway by knocking out or knocking down the key players of the IFN response, and for the first time, we conclusively demonstrate that RIG-I and MAVS but not STING, are essential to promote A3A expression after viral infection. Moreover, we are also the first to show that MDA5 has a backup function to induce A3A expression when RIG-I is not present. These results significantly advance our fundamental understanding of A3A regulation by the IFN response and establish the tools to address the second goal of our study.

We again agree with the reviewer that the activation of JAK/STAT signaling pathways by IFN is known to induce some members of the APOBEC family. However, it was still unknown whether the JAK/STATs pathways induce A3A expression directly or through the regulation of an ISG that will in turn regulate A3A. Moreover, it was still not known which STAT transcription factor(s) regulate A3A, where STAT transcription factor(s) were recruited on the A3A promoter, or whether JAK inhibitors could be used to efficiently suppress A3A expression in cancer cells. Here for the first time, we identify that STAT2 is essential for A3A expression during the IFN response. We demonstrate that STAT2 is directly recruited to the A3A promoter, and we identify where STAT2 is recruited on the A3A promoter. Finally, we show that the JAK inhibitors completely abrogate 3p-hpRNA induced A3A level, highlighting JAK inhibitors as promising candidates in the clinic to suppress A3A-induced mutation in patients' tumors. Together, these new results further enhance our fundamental knowledge on A3A regulation by the IFN

response. More importantly, *the first part of our study provides the tools to address our second goal to identify other stress(es) that induce A3A independently of the interferon response.* Indeed, RIG-I and STAT2 KO cells we used to study the IFN-response regulated A3A provided us important tools to identify additional pathways regulating A3A expression independently of the IFN response.

The use of RIG-I and STAT2 KO cell lines was particularly important for the interpretation of our transcriptomic analysis and allowed us to identify and separate two populations of ISGs. The first ISG population is strictly regulated by the IFN response, while the second population is regulated by both the IFN response and the NF κ B pathway, depending on the type of stress. This is an important new concept in the field since we are now proposing that the expression of many ISGs in cells is not necessarily related to the activation of the interferon response. The use of RIG-I and STAT2 KO cell lines was crucial to establish that after DNA damage and replication stress, a population of ISGs is up-regulated independently of the IFN response. Our study goes beyond A3A regulation, and we now propose that two distinct mechanisms regulate a whole population of inflammatory genes. These results have strong clinical implications since ISGs levels are often monitored in patients to determine treatment. Our results suggest that it is important to look at specific ISGs in order to determine correctly which pathway is activated. For example, as we show in our study, JAK inhibitors that are currently used clinically would very efficiently suppress IFN response-induced A3A in patients but be completely inefficient at inhibiting DNA damage-induced A3A.

Together, the first part of our manuscript presents new important results and concepts that will enhance our fundamental understanding of A3A regulation in cells and have direct clinical implications. We sincerely thank the reviewer for helping us to explain better and highlight the novelty of our work. We have now modified our manuscript accordingly.

It was interesting to note that DNA damage-induced A3A expression is not dependent on RIG MAVS signaling or JAK/STAT signaling in cancer cells and p65/NF κ B play important role in the regulation of A3A during DNA damage in cancer cells. Again, it has been shown previously in papers published in Cancer Research and BBRC journals that p65 or RELB binds in A3B promoter and important for transcription of A3B in cancer cells (PMID: 27577680; PMID: 26420215). So it's not surprising that A3A is also controlled by NF κ B where it is very well known that DDR induces NF κ B signaling (PMID: 28626800). There is some new information here (A3A vs A3B) but I doubt that this is enough to get in this journal.

In nutshell, there is no doubt that the study is performed very nicely. However, the knowledge that comes from this manuscript is incremental.

We would like to thank the reviewer's acknowledgment of the interesting aspects of our work as well as the quality of the data generated by a graduate student in my lab. In the second part of our study, it was indeed very surprising and unexpected for us to find that DNA damage-induced A3A expression occurs through a completely independent mechanism that does not require the IFN response (please see the final comment for more information about this specific point). While the activation of the NF κ B pathway by DDR has been well documented, it is still not well characterized which genes are regulated after different types of DNA damage and what are the consequences of this activation, justifying a deeper characterization. Below we will provide evidences of how different types of DNA damage lead to different cell responses and gene expression patterns.

As the reviewer stated, several mechanisms regulating A3B expression have been characterized. However, previous observations suggest that the mechanisms governing A3A expression are different from those regulating A3B. The most important piece of evidence of this differential regulation between A3A and A3B expression is revealed when we look directly at patients' tumors. Both A3A and A3B are overexpressed in many tumors, but not necessarily in the same tumor types, and it is still not understood why A3A and A3B are expressed in different types of cancers. For example, the Harris lab demonstrated that A3B is often overexpressed in ovarian cancers but not A3A (Leonard et al. *Cancer Research*, 2013), suggesting that mechanisms regulating A3B expression are independent of those regulating A3A. In addition, we recently demonstrated that A3A and A3B-induced mutations accumulate in different types of tumors (Jalili et al. *Nature Communications*, 2020), further suggesting that A3A and A3B are regulated through different mechanisms.

To further support that the mechanisms regulating A3A and A3B are not the same and what we know about A3B regulation cannot be directly applied to A3A, we have now performed numerous additional experiments to compare A3A and A3B expression following different types of stresses. We first show that viral infection strongly induces A3A expression but not A3B (**new Supplementary Figure 2G**). We now show that different types of DNA damage lead to a differential regulation of A3A or A3B expression. On one hand, cells treated with IR+ATRi induce A3A expression but not A3B (**Rebuttal Figure 2A**). On the other hand, cells treated with topoisomerase inhibitors (CPT, TPT, or ETP) induce A3B expression but not A3A (**Rebuttal Figures 2B, 2C, and 2D**). Finally, DNA replication stress (HU treatment) stimulates both A3A and A3B expression, but the addition of ATR inhibitor results in an increase of A3A expression and a decrease in A3B expression (**Rebuttal Figures 2E and 2F**). Together, these results suggest that A3A and A3B are regulated through different mechanisms and are not necessarily activated after the same type of DNA damage. In this study, we demonstrated for the first time how the canonical NF κ B pathway through p65 regulates A3A after different types of DNA damage. However, it is still unclear how A3B expression is regulated during the DNA damage response. We are currently working on dissecting those mechanisms. As such, we would prefer to save these results for a future manuscript dedicated to A3B regulation by the DNA damage response.

As mentioned by the reviewer, two previous studies found that A3B expression is induced after PKC induction, but these two studies had conflicting conclusions about which sub-NF κ B pathway regulates A3B (Leonard et al. *Cancer Research*, 2015 / Maruyama et al. *BBRC*, 2016). While we do not wish to speculate on how A3B is regulated following PKC induction (a very different type of stress), our new data strongly support that A3A and A3B are regulated through different mechanisms after different types of DNA damage (**Rebuttal Figure 2**). Studies from the Harris and Ali labs suggested that RELB, p53, and E2F4/6 are important factors regulating A3B and these factors may play an essential role in the regulation of A3B during the DNA damage response (Leonard et al. *Cancer Research*, 2015 / Periyasamy et al. *NAR*, 2017 / Roelofs et al. *eLife*, 2020). However, further studies are necessary to determine whether those factors affect A3B expression during the DNA damage expression especially following treatment with topoisomerase inhibitors.

Minor Points

1. Why author did not check the role of TLR3 or IF16 in the regulation of APOBEC3A?

This is a very important point. We did check the role of TLR3 and IFI16 during the initiation of this project and we did not observe any defect of A3A expression in their absence after viral infection (**Rebuttal Figure 3A and 3B**). These data are consistent with the results shown in **Figure 1A and Supplementary Figure 1A** where we transfected cells with VACV-70 and poly(A:U) oligonucleotides known to stimulate IFI16 and TLR3, respectively (Unterholzner et al. Nature Immunology 2010 / Perrot et al. J. Immunology 2010 / Conforti et al. Cancer research 2010). Neither VACV-70 nor poly(A:U) transfection triggers A3A expression, further suggesting that neither TLR3 nor IFI16 regulate A3A expression. Finally, we show that the knockout of MAVS completely abrogates the upregulation of A3A after poly(I:C), which is another RNA substrate known to stimulate TLR3 in addition to RIG-I/MDA5/MAVS (**Supplementary Figure 2B**). This result further suggests that poly(I:C) induces A3A through the activation of the RIG-I/MDA5/MAVS pathway and not TLR3. If the reviewer thinks it essential, we will be happy to include these new panels as well as the related IFI16 and TLR3 control knockdown experiments in the revised manuscript.

Figure Rebuttal 3: A-B. BICR6 cells were transfected with TLR3 or IFI16 siRNA for 36h following by SeV infection (1 MOI) for 16h. A3A expression level was determined by RT-qPCR. Error bar: S.D. (n = 3).

2. Both the introduction and discussion have too much information and very extensive review of the literature, several of the things are not very relevant to the manuscript. This can be reduced.

We agree with the reviewer that our introduction and discussion are extensive. It was important for us to fully acknowledge and cite previously published work from other labs that allowed us to advance this project. We hope our introduction will prove helpful for the reader to have a comprehensive overview of the topic since this study focuses on many different pathways. We have now removed several sections in our discussion and if the reviewer thinks additional sections are ancillary to this story, we will remove them.

3. This is surprising that DNA damage-induced expression of A3A is not dependent on IFN response. Previously, it has been shown that DNA damage induces IFN response via cGAS-STING pathways (PMID: 28738408 PMID: 22013119 PMID: 25692705). More specifically, a manuscript published in the NAR journal (PMID: 28100701) showed that “cytoplasmic DNA triggers interferon α and β production through the RNA polymerase III transcription/RIG-I pathway leading to massive upregulation of APOBEC3A”. Is it possible that the model used in your manuscript to cause DNA Damage (HU+ATRi) is different from previous works? This discrepancy should be resolved.

It was indeed very surprising and unexpected for us to find that DNA damage-induced A3A expression was through a completely independent mechanism that does not require the IFN response. *This result is a major novelty of our study.* As mentioned by the reviewer, current models suggest that ISGs are induced after DNA damage through the activation of the IFN response by the activation of cGAS/STING or RIG-I/MAVS pathways (Reislander et al. Molecular Cell, 2020). Here, we add a new central concept to this model where a sub-population of ISGs, including A3A, is regulated through an IFN independent mechanism after DNA damage that does not require either cGAS/STING or RIG-I/MAVS but instead are regulated by the canonical NFKB pathway. This is an important new model for the DNA damage field since it was previously assumed that all ISGs are regulated through the IFN response after DNA damage.

As mentioned by the reviewer, a previous study reported that cytoplasmic mitochondrial DNA directly transfected into cells triggers an IFN response leading to the upregulation of A3A. The results reported in that paper are in complete agreement with our model since we also confirmed that other types of cytoplasmic DNA (polydA:dT), like dsRNA, stimulates A3A expression through the IFN response (**Supplementary Figure 2A**). However, above-mentioned paper did not look at whether DNA damage modulates A3A expression through a similar mechanism. When we monitored the level of IFN response after DNA damage, we found that the activation of the IFN response is very weak compared to the activation of the IFN response after transfection of exogenous nucleic acids in the cytoplasm, while the level of induced A3A is similar between both stresses (**Figure 5C, 5D, 5J, and 5K**). This result suggests that DNA damage does not generate as much endogenous cytosolic DNA compared to a high amount of exogenous nucleic acids directly transfected into cells. This observation is also consistent with a recent study showing that after DNA damage the IFN production is counteracted in part by TREX1 digestion of cytosolic DNA to inhibit cGAS activation (Mohr et al. Molecular Cell, 2021). This unexpected observation is what has led us to investigate for another mechanism that might regulate A3A expression after DNA damage. We now show that A3A is regulated through an independent mechanism after genotoxic stress and does not require the IFN response.

Altogether, our study for the first time demonstrated that there are *two fully independent pathways* that modulate A3A expression in cells, depending of the type of stress. Surprisingly, we found that DNA damage-induced IFN response via cGAS-STING pathways is not sufficient to triggers a high level of A3A expression and cells instead utilize the NFκB pathway to efficiently induce A3A and other inflammatory genes.

REVIEWERS' COMMENTS

Reviewer #1 (Remarks to the Author):

The authors have gone to significant lengths to address my previous comments and I am happy that they have been able to do so. I don't have any further concerns.

Tim Fenton

Reviewer #2 (Remarks to the Author):

The authors have addressed all of my concerns and I am supportive of publication

Reviewer #3 (Remarks to the Author):

The authors have clarified some important concerns raised regarding the novelty of the work. I am in favor of the publication of this manuscript. For the reason that work is performed systematically to understand the regulation of A3A and a large amount of convincing data is provided. I hope the mechanistic understanding (which is not very clear in this manuscript) may come in a follow-up manuscript.

REVIEWER COMMENTS

REVIEWERS' COMMENTS

Reviewer #1 (Remarks to the Author):

The authors have gone to significant lengths to address my previous comments and I am happy that they have been able to do so. I don't have any further concerns.

Tim Fenton

We thank the reviewer for his/her appreciation of the significance and quality of our work.

Reviewer #2 (Remarks to the Author):

The authors have addressed all of my concerns and I am supportive of publication

We thank the reviewer for his/her appreciation of our work.

Reviewer #3 (Remarks to the Author):

The authors have clarified some important concerns raised regarding the novelty of the work. I am in favor of the publication of this manuscript. For the reason that work is performed systematically to understand the regulation of A3A and a large amount of convincing data is provided. I hope the mechanistic understanding (which is not very clear in this manuscript) may come in a follow-up manuscript.

We thank the reviewer for his/her support.